# DEEP GENERATIVE PRIORS FOR 3D BRAIN ANALYSIS

## ABSTRACT

Diffusion models have recently emerged as powerful generative models in medical imaging. However, it remains a major challenge to combine these data-driven models with domain knowledge to guide brain imaging problems. In neuroimaging, Bayesian inverse problems have long provided a successful framework for inference tasks, where incorporating domain knowledge of the imaging process enables robust performance without requiring extensive training data. However, the anatomical modeling component of these approaches typically relies on classical mathematical priors that often fail to capture the complex structure of brain anatomy. In this work, we present the first general-purpose application of diffusion models as priors for solving a wide range of medical imaging inverse problems. Our approach leverages a score-based diffusion prior trained extensively on diverse brain MRI data, paired with flexible forward models that capture common image processing tasks such as super-resolution, bias field correction, inpainting, and combinations thereof. We further demonstrate how our framework can refine outputs from existing deep learning methods to improve anatomical fidelity. Experiments on heterogeneous clinical and research MRI data show that our method achieves state-of-the-art performance producing consistent, high-quality solutions without requiring paired training datasets. These results highlight the potential of diffusion priors as versatile tools for brain MRI analysis.

## 1 INTRODUCTION

Magnetic resonance imaging (MRI) stands as one of the most versatile and informative neuroimaging modalities, providing detailed insights into the living brain. However, a substantial portion of the vast amounts of human brain MRI data collected worldwide remains underutilized due to acquisition limitations that result in images that are unsuitable for most downstream tasks. Most neuroimaging analysis tools, e.g., SPM Ashburner & Friston (2005), FSL Jenkinson et al. (2012) and FreeSurfer Fischl (2012), assume access to high-resolution, 1 mm isotropic scans across standardized contrasts Kofler et al. (2024); Blumenthal et al. (2002); Klapwijk et al. (2019); Iglesias et al. (2021). However, acquiring such scans is costly, requiring longer scan times and higher field strengths. Moreover, these methods assume a level of homogeneity that is rarely present in clinical practice, where variability may arise from both acquisition factors (such as choice of contrast, anisotropy, motion-corrupted slices, or low field strength) and biological differences (including normal anatomical variation and pathological effects). Furthermore, ultra low-field MRI has emerged as a promising low-cost and portable alternative to high-field MRI Sorby-Adams et al. (2024). However, the low signal-to-noise ratio and spatial resolution currently limits its applicability.

This disparity between ideal and available data has motivated extensive research into image enhancement methods. Bayesian inverse problems have long been a popular approach, leveraging well-established forward models to provide principled solutions grounded in domain knowledge of the imaging process Balbastre et al. (2018); Brudfors et al. (2019). However, although the likelihood models represent the underlying physics well, these approaches typically rely on classical mathematical priors that are insufficient to capture the complex anatomical structures characteristic of brain imaging data.

Conversely, modern deep learning approaches can learn sophisticated image statistics from large datasets Islam et al. (2023); Safari et al. (2025); Bercea et al. (2024). However, they often neglect crucial domain knowledge about the underlying problem and rely on paired training data, which is frequently unavailable. As a result, researchers must either train on small datasets that may not gen-

eralize well or use synthetic data Kalluvila et al. (2022); Lawry Aguila et al. (2025) that may fail to bridge the domain gap when applied to real-world datasets. Moreover, most data-driven methods are designed for specific processing tasks or imaging modalities, limiting their generalizability across the diverse range of problems encountered in clinical practice.

Recently, diffusion models have emerged as a powerful class of generative models, demonstrating remarkable success in medical imaging applications including synthesis Pinaya et al. (2022), segmentation Fernandez et al. (2022), and anomaly detection Lawry Aguila et al. (2025); Wolleb et al. (2022). In computational imaging more broadly, researchers have begun combining diffusion model priors with explicit forward models to solve inverse problems Chung et al. (2023); Kawar et al. (2022); Zhang et al. (2024), enabling principled solutions that leverage both powerful generative models and task-specific domain knowledge. This promising framework remains largely unexplored in neuroimaging, where the complex anatomy and diverse imaging challenges presents a unique opportunity for data-driven inverse problem solving.

In this work, we present the first general-purpose application of diffusion models as priors for solving medical imaging inverse problems. Our approach combines a score-based diffusion prior, trained on a large and diverse brain MRI cohort, with flexible forward models that can handle a wide range of imaging scenarios. Unlike existing data-driven methods that require paired training data for each specific task, our framework operates by solving inverse problems directly, making it highly versatile and applicable to scenarios where paired training data may not exist.

Our key contributions include: *(i)* A unified probabilistic framework for brain MRI analysis that combines powerful data-driven diffusion priors with knowledge-based forward models. *(ii)* A range of likelihood formulations designed to address a number of challenges in the medical imaging field, including; super-resolution, bias field correction, disease inpainting, and image enhancement. *(iii)* We demonstrate the robustness and versatility of our method by applying it to a range of challenging heterogeneous datasets, including real-world clinical and ultra low-field data, showing that it can consistently generate high-quality images and outperform baseline approaches.

## 2 BACKGROUND

### 2.1 INVERSE PROBLEMS IN MEDICAL IMAGING

Many tasks in medical imaging can be formulated as inverse problems, where we seek to recover an unknown image $\mathbf{x} \in \mathbb{R}^{d_x}$ from observed measurements $\mathbf{y} \in \mathbb{R}^{d_y}$ related by:

$$\mathbf{y} = F(\mathbf{x}) + \epsilon \tag{1}$$

where the forward model $F$ is assumed to be well established and $\epsilon$ is the measurement noise. When $\mathbf{y}$ provides incomplete information about $\mathbf{x}$ then solving for $\mathbf{x}$ is ill-posed. The Bayesian framework addresses this by introducing a prior that encodes assumptions about plausible solutions. The inverse problem is then expressed through the posterior distribution:

$$\log p(\mathbf{x} \mid \mathbf{y}) = \log p(\mathbf{y} \mid \mathbf{x}) + \log p(\mathbf{x}) + \text{const} \tag{2}$$

which naturally decomposes inference into a data-fitting term (likelihood) and a regularizer (prior). Traditionally in medical imaging, regularizers which enforce some property of an image such as smoothness Ehrhardt & Betcke (2016); Brudfors et al. (2019) or sparsity Lustig et al. (2007); Arridge (1999) are used as priors. However, these priors fail to capture the complex structure of the brain.

### 2.2 SCORE-BASED DIFFUSION MODELS

Diffusion models Ho et al. (2020); Song et al. (2021) define a forward stochastic process that gradually transforms data samples $\mathbf{x}_0 \sim q(\mathbf{x}_0)$ into samples from a known prior distribution $p_T(\mathbf{x})$, which is generally Gaussian. This transformation is achieved via a time-indexed sequence of variables $\{\mathbf{x}_t\}_{t=0}^{T}$ governed by a linear stochastic differential equation (SDE):

$$d\mathbf{x}_t = \mathbf{f}(\mathbf{x}_t, t)dt + g(t)d\mathbf{w}_t, \tag{3}$$

where $\mathbf{f} : \mathbb{R}^d \times [0, T] \to \mathbb{R}^d$ is the drift function, $g : [0, T] \to \mathbb{R}$ is the diffusion coefficient, and $\mathbf{w}_t$ is a Wiener process. Using the EDM framework Karras et al. (2022), the transition kernel $p(\mathbf{x}_t \mid$

$\mathbf{x}_0) \sim \mathcal{N}(\mathbf{x}_0, \sigma_t^2 \mathbf{I})$ is a Gaussian with parameters controlled by $\sigma_t$, a predefined noise schedule. These choices in the forward process ensure that the terminal distribution is approximately Gaussian $p_T(\mathbf{x}) \approx \mathcal{N}(0, \sigma_T^2 \mathbf{I})$.

To sample from $q(\mathbf{x}_0)$, we can solve the reverse-time SDE:

$$dx_t = \left[ \mathbf{f}(\mathbf{x}_t, t) - g^2(t) \nabla_{\mathbf{x}_t} \log p(\mathbf{x}_t) \right] dt + g(t) \, d\hat{\mathbf{w}}_t, \tag{4}$$

which shares the same marginals $\{p(\mathbf{x}_t)\}_{t=0}^{T}$ as the forward process and $d\hat{\mathbf{w}}_t$ is the reverse-time Weiner process. The score function $\nabla_{\mathbf{x}_t} \log p(\mathbf{x}_t)$ can be approximated using a neural network $\mathbf{s}_\theta$, trained via the denoising score matching objective:

$$\mathcal{L}(\theta) = \mathbb{E}_{\mathbf{x}_t \sim p(\mathbf{x}_t | \mathbf{x}_0), \, \mathbf{x}_0 \sim q(\mathbf{x}_0), \, t \sim \mathcal{U}(0,T)} \left[ \left\| \mathbf{s}_\theta(\mathbf{x}_t, t) - \nabla_{\mathbf{x}_t} \log p(\mathbf{x}_t \mid \mathbf{x}_0) \right\|^2 \right] \tag{5}$$

which is tractable because the transition kernel has known mean and variance from the forward SDE.

### 2.3 POSTERIOR SAMPLING FOR INVERSE PROBLEMS

Score-based diffusion models can serve as powerful learned priors $p(\mathbf{x})$ for inverse problems by leveraging their ability to capture complex data distributions. Instead of classical regularizers, we can use the score function $\mathbf{s}_\theta(\mathbf{x}_t, t) \approx \nabla_{\mathbf{x}_t} \log p(\mathbf{x}_t)$ as a data-driven prior that has learned realistic image statistics from large datasets. By leveraging the diffusion model as a prior, it is possible to modify Equation 4 such that the reverse SDE for sampling from the posterior distribution becomes Chung et al. (2023):

$$dx_t = \left[ \mathbf{f}(\mathbf{x}_t, t) - g^2(t)(\nabla_{\mathbf{x}_t} \log p(\mathbf{y} \mid \mathbf{x}_t) + \nabla_{\mathbf{x}_t} \log p(\mathbf{x}_t)) \right] dt + g(t) \, d\hat{\mathbf{w}}_t. \tag{6}$$

While the prior gradient $\nabla_{\mathbf{x}_t} \log p(\mathbf{x}_t)$ is readily available from the pre-trained score network, the true likelihood gradient requires computing:

$$\nabla_{\mathbf{x}_t} \log p(\mathbf{y} \mid \mathbf{x}_t) = \nabla_{\mathbf{x}_t} \log \int p(\mathbf{y} \mid \mathbf{x}_0) \, p(\mathbf{x}_0 \mid \mathbf{x}_t) \, d\mathbf{x}_0 \tag{7}$$

which is intractable because it involves integrating over all possible clean images $\mathbf{x}_0$ that could have generated the noisy diffusion state $\mathbf{x}_t$.

Due to this intractability, researchers have introduced several strategies to approximate the noisy likelihood and enable posterior sampling Chung et al. (2023); Kawar et al. (2022); Zhang et al. (2024); Feng et al. (2023); Wang et al. (2023); Dou & Song (2024). These advances have facilitated the real-world application of diffusion priors for solving inverse problems Zheng et al. (2025). In medical imaging, these approaches have been used for image reconstruction of MRI Jalal et al. (2021); Song et al. (2022), where $\mathbf{y}$ corresponds to k-space measurements (spatial frequencies in the Fourier domain), and CT Chung et al. (2022); Song et al. (2022), where $\mathbf{y}$ corresponds to sinograms generated from X-ray projections at multiple angles. Importantly, however, our approach differs from these existing methods, which require incorporating acquisition measurements into the likelihood. Instead, our method operates in the image space such that it can be applied to scenarios where acquisition parameters are not available, as is the case in most clinical settings and archived datasets. Our work is also related to the recent study by Kim et al. (2025), which adapts the diffusion posterior sampling (DPS) approach Chung et al. (2023) to reduce hallucinations in super-resolved images of low-resolution MRI generated by deep generative models. In this work, we adopt a generic, task-agnostic approach to medical imaging challenges, introducing a versatile framework that can be applied across modalities and datasets without requiring task-specific training.

## 3 DIFFUSION PRIORS FOR MEDICAL IMAGING PROBLEMS

### 3.1 A PRIOR FOR THE BRAIN

The first step in our medical imaging inverse problem framework is to train a data-driven prior. This prior should be trained on images that are both high-quality and representative of the target distribution we wish to recover through our inverse problem solver. In medical imaging, it is often desirable to obtain a high-resolution (1 mm isotropic) scan of the brain, with many neuroimaging

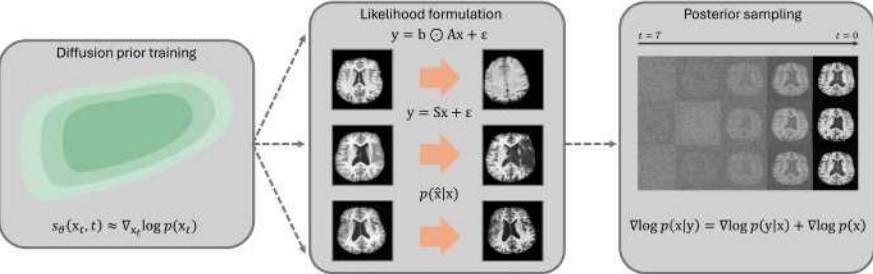

Figure 1: Overview of our approach to use diffusion priors for inverse problems in 3D brain analysis. (Left) Training phase learns the diffusion prior score function from diverse brain data. (Middle) Task-specific likelihood formulations for different medical imaging problems. (Right) DAPS algorithm samples from posterior distribution to generate clean images.

software packages tailored for such data Ashburner & Friston (2005); Jenkinson et al. (2012); Fischl (2012). Furthermore, large pathological structures, such as brain tumours, can cause these tools to fail Kofler et al. (2024), making the analysis of disease effects and clinical decision-making more difficult. A prior trained on healthy brain anatomy could enable super-resolution and restoration of low-resolution scans and inpainting of pathological regions, allowing the application of standard neuroimaging pipelines. We therefore assemble a large cohort of 1 mm isotropic images of healthy subjects spanning multiple public datasets, contrasts, and demographics (described in Section 4.1.1). This diverse cohort is designed to minimize the domain gap between the prior and target images and capture the substantial variability present in MRI data. Importantly, this dataset consists of artifact-free, healthy, high-resolution scans and thus any corruption is not learned by the prior but is instead explicitly modeled within our inverse problem formulation.

### 3.2 POSTERIOR SAMPLING

To sample from the posterior distribution, we take the approach proposed by Zhang et al. (2025) where they approximate the likelihood gradient in Equation 6 by introducing a decoupled noise annealing process to consecutively sample from $p(\mathbf{x}_t \mid \mathbf{y})$. At each step, we first draw an approximate clean sample $\mathbf{x}_{0|y} \sim p(\mathbf{x}_0|\mathbf{x}_{t+\Delta t}, \mathbf{y})$, and then reapply the forward diffusion kernel to obtain $\mathbf{x}_t \sim \mathcal{N}(\mathbf{x}_{0|y}, \sigma_t^2 \mathbf{I})$. To sample $\mathbf{x}_{0|y}$, we apply the Langevin dynamics Welling & Teh (2011) update rule given by:

$$\mathbf{x}_0^{(j+1)} \leftarrow \mathbf{x}_0^{(j)} + \eta_t \left( \nabla_{\mathbf{x}_0^{(j)}} \log p(\mathbf{x}_0^{(j)} \mid \mathbf{x}_t) + \nabla_{\mathbf{x}_0^{(j)}} \log p(\mathbf{y} \mid \mathbf{x}_0^{(j)}) \right) + \sqrt{2\eta_t}\, \epsilon_j, \quad \epsilon_j \sim \mathcal{N}(0, I) \quad (8)$$

where $\eta_t$ is the step size at time $t$. We can approximate the conditional distribution $p(\mathbf{x}_0|\mathbf{x}_t) \approx \mathcal{N}(\mathbf{x}_0; \mathbf{x}_\theta(\mathbf{x}_t, t), \tau_t^2 \mathbf{I})$ where $\mathbf{x}_\theta(\mathbf{x}_t, t)$ is the predicted denoised data at time $t = 0$ predicted by the diffusion model and the variance $\tau_t^2$ is specified heuristically.

### 3.3 LIKELIHOOD FORMULATION FOR MEDICAL IMAGING PROBLEMS

Whilst a high-resolution scan is often desirable for a number of medical imaging tasks, in practice, however, often only partial or degraded information is available, for example, a lower-resolution image, an image with corrupt slices, or with pathology. These challenging scenarios can be naturally formulated as inverse problems. Let $\mathbf{x}$ denote the unknown high resolution image and $\mathbf{y}$ denote the observed image. Assuming the noise $\epsilon \sim \mathcal{N}(0, \tau^{-1}\mathbf{I})$ is Gaussian with precision $\tau$, we can use Equation 1 to introduce a general likelihood for common medical imaging tasks:

$$\mathbf{y} \mid \mathbf{x} \sim \mathcal{N}\big(\mathbf{y} \mid F(\mathbf{x}; \theta), \tau^{-1}\mathbf{I}\big), \quad p(\mathbf{y} \mid \mathbf{x}) \propto \exp\big(-\tfrac{\tau}{2} \|\mathbf{y} - F(\mathbf{x}; \theta)\|_2^2\big). \quad (9)$$

Here, $F(\mathbf{x}; \theta)$ represents the forward operator that maps the high-resolution image to the measurement space, $\theta$ denotes problem-specific parameters that may be optimized jointly with the reconstruction, and $\tau$ accounts for both acquisition noise and model uncertainty. We apply this inverse problem framework to three key challenges in medical imaging: image restoration for generating high-quality 1 mm isotropic images from acquisitions with lower resolution, inpainting of pathological tissue, and image refinement for enhancing the results of existing image processing methods.

**Image restoration.** For image super-resolution or restoration tasks, we need to consider the following elements in our forward model; resolution modeling, image alignment, and bias field correction. The first two points can be addressed by considering a deterministic projection matrix $\mathbf{A}$, well established in the MRI super-resolution literature Balbastre et al. (2018); Brudfors et al. (2019), as a sequence of linear operators, $\mathbf{A} = \mathbf{RST}$. First, the $\mathbf{T}$ operator aligns the high resolution image $\mathbf{x}$ to the low resolution image $\mathbf{y}$ field-of-view. Secondly, the $\mathbf{S}$ operator simulates the slice profile of MRI acquisition, functioning as an anisotropic blurring operator. Following previous work Brudfors et al. (2019), we assume a Gaussian slice profile and infer the slice gap using the image metadata. Finally, the $\mathbf{R}$ operator performs downsampling to the low-resolution grid.

The second aspect to consider is removing bias field effects. Clinical MRI images are corrupted by spatially-varying intensity inhomogeneities known as bias fields, which arise from imperfections in the RF coils and $B_0$ field variations Van Leemput et al. (1999). The bias field is smooth and multiplicative in nature, meaning that the observed intensity at each voxel is the product of the true tissue intensity and a spatially-varying multiplicative factor. We model the bias field $\mathbf{b}$ as a vector where each element is defined by:

$$b_i = \exp\Big( \sum_k c_k \, \phi_k(r_i) \Big). \tag{10}$$

where $\phi_k(r_i)$ are smooth basis functions evaluated at spatial location $r_i$ and $c_k$ are the corresponding coefficients. In our implementation, we use 3rd order polynomial basis functions and initialize $\mathbf{c}$ using the N4ITK algorithm Tustison et al. (2010). As in previous work Ashburner & Friston (2005); Cerri et al. (2023), we use a smoothing prior $p(\mathbf{c}) \propto \exp(-\lambda\|\mathbf{c}\|^2)$ where $\lambda$ is chosen heuristically. Combining these elements we define the following likelihood:

$$\mathbf{y} \mid \mathbf{x}, \mathbf{c} \sim \mathcal{N}\big(\mathbf{y} \mid (\mathbf{b} \odot \mathbf{A}\mathbf{x}), \tau^{-1}\mathbf{I}\big), \quad p(\mathbf{y} \mid \mathbf{x}, \mathbf{c}) = \frac{\tau^{N/2}}{(2\pi)^{N/2}} \exp\Big( -\frac{\tau}{2}\|\mathbf{y} - \mathbf{b} \odot \mathbf{A}\mathbf{x}\|_2^2 \Big). \tag{11}$$

To optimize both $\mathbf{x}$ and $\mathbf{c}$, we perform alternate updates via coordinate descent. For gaussian observation noise $\mathbf{n} \sim \mathcal{N}(0, \tau_y^{-1}\mathbf{I})$, the $\mathbf{x}$ update rule in Equation 8 simplifies to:

$$\mathbf{x}_0^{(j+1)} = \mathbf{x}_0^{(j)} - \eta \nabla_{\mathbf{x}_0^{(j)}} \frac{\|\mathbf{x}_0^{(j)} - \hat{\mathbf{x}}_0(\mathbf{x}_t)\|^2}{2\tau_t^2} - \eta \nabla_{\mathbf{x}_0^{(j)}} \frac{\|\mathbf{b} \odot \mathbf{A}\mathbf{x}_0^{(j)} - \mathbf{y}\|^2}{2\tau_y^2} + \sqrt{2\eta}\epsilon_j \tag{12}$$

Given that $\log p(\mathbf{c}|\mathbf{x}, \mathbf{y}) = \log p(\mathbf{y}|\mathbf{c}, \mathbf{x}) + \log p(\mathbf{c}) + \text{const}$, we can define the $\mathbf{c}$ update as:

$$\mathbf{c}^{(k+1)} = \mathbf{c}^{(k)} - \alpha(t) \left[ \nabla_{\mathbf{c}} \left( \frac{\|\mathbf{y} - \mathbf{b} \odot \mathbf{A}\mathbf{x}_0^{(j)}\|^2}{2\tau_y^2} + \frac{\lambda\|\mathbf{c}\|^2}{2} \right) \right] \tag{13}$$

where $\alpha(t)$ is an annealing schedule that scales the bias field update based on the diffusion timestep, providing smaller updates early in the reverse process when $\mathbf{x}_0^{(j)}$ is noisy and larger updates as the image estimate becomes more reliable.

**Inpainting.** In some cases, we may wish to inpaint disease or corrupt regions of an image with realistic healthy tissue while preserving individual anatomical characteristics. This enables the use of a wide range of existing analysis tools that often fail or produce unreliable results in the presence of pathology. For such inpainting tasks, we can define the likelihood given a binary mask $\mathbf{m} \in \{0,1\}^M$ where $m_i = 1$ indicates healthy pixels and $m_i = 0$ indicates pathology pixels and defining a selection matrix $\mathbf{S} \in \{0,1\}^{N \times M}$ where $N$ is the number of healthy pixels:

$$\hat{\mathbf{y}} \mid \mathbf{x}, \mathbf{m} \sim \mathcal{N}\big(\hat{\mathbf{y}} \mid \mathbf{S}\mathbf{x}, \tau_y^{-1}\mathbf{I}\big) \tag{14}$$

where $\hat{\mathbf{y}} = \mathbf{S}\mathbf{y}$ represents the observed healthy pixels extracted from the full observation $\mathbf{y}$. The update becomes:

$$\mathbf{x}_0^{(j+1)} = \mathbf{x}_0^{(j)} - \eta \nabla_{\mathbf{x}_0^{(j)}} \frac{\|\mathbf{x}_0^{(j)} - \hat{\mathbf{x}}_0(\mathbf{x}_t)\|^2}{2\tau_t^2} - \eta \nabla_{\mathbf{x}_0^{(j)}} \frac{\|\mathbf{S}\mathbf{x}_0^{(j)} - \hat{\mathbf{y}}\|^2}{2\tau_y^2} + \sqrt{2\eta}\epsilon_j \tag{15}$$

This formulation allows the prior to determine the values of disease regions while constraining healthy pixels to match the data.

**Image refinement.** Many existing image processing tools provide approximate solutions that could benefit from further refinement. While these methods have proven valuable for processing heterogeneous data, their outputs often exhibit characteristic artifacts such as over-smoothing of fine details or inconsistencies with the underlying morphology. For example, SynthSR Iglesias et al. (2023) can fail to fully inpaint pathology or smooths images. Our diffusion-based inverse problem framework provides a principled approach to refine outputs from any existing method by treating them as initial approximations that can be iteratively improved. We formulate this as a constrained reconstruction problem where we seek to generate a high-quality image $\mathbf{x}$ that maintains consistency with the initial approximation $\hat{\mathbf{x}}$ from the existing method. We construct the likelihood and posterior:

$$\hat{\mathbf{x}} \mid \mathbf{x} \sim \mathcal{N}(\hat{\mathbf{x}} \mid \mathbf{x}, \tau_s^{-1}\mathbf{I}), \quad \log p(\mathbf{x} \mid \hat{\mathbf{x}}) = \log p(\hat{\mathbf{x}} \mid \mathbf{x}) + \log p(\mathbf{x}) + \text{const.} \tag{16}$$

where $\tau_s$ controls the trust placed in the initial approximation.

## 4 EXPERIMENTS

In this section, we evaluate the performance of our method on three medical imaging inverse problem tasks; image restoration, image inpainting and image refinement. For the image restoration and image inpainting tasks, we compare our method against a number of both traditional and data driven baselines. For the image refinement task, we qualitatively assess the ability of our method to improve the quality of image generated by SynthSR Iglesias et al. (2023), a machine learning method for joint super-resolution and anomaly inpainting of T1w MRI brain scans.

### 4.1 EXPERIMENTAL SETUP

We use a U-net Dhariwal & Nichol (2021) for our diffusion model prior backbone with an image resolution of 176×176×176 and 128 model channels. The network uses channel multipliers of [1, 2, 2] with a channel embedding multiplier of 4, and includes attention at resolution 16 with one block per resolution and a single attention head. We train the model using the EDM framework Karras et al. (2022), with noise levels sampled from a log-normal distribution (mean -0.5, standard deviation 1.5) and data standard deviation of 0.5. We use the Adam optimizer with a learning rate of $1 \times 10^{-4}$ and train for 500,000 steps (including 500 warmup steps) with gradient clipping at 1.0. To stabilize training, we maintain an exponential moving average of the weights with decay 0.9999, updated every 10 steps. We use the DAPS Zhang et al. (2025) algorithm for posterior sampling with the parameters given in table 1. The parameters are chosen heuristically using the synthetic data described in Appendix A.4 and based on prior work Zheng et al. (2025). The $\tau$ values are described in the Section 4.5.

Table 1: DAPS Algorithm Parameters

| Annealing Steps | Annealing $\sigma_{\max}$ | Annealing $\sigma_{\min}$ | Diffusion Steps | Diffusion $\sigma_{\min}$ | Langevin Step Size | Langevin Step No. | Noise ($\tau$) | Decay Ratio | Schedule | Timestep |
|---|---|---|---|---|---|---|---|---|---|---|
| 50 | 100 | 0.1 | 5 | 0.01 | $1 \times 10^{-4}$ | 20 | – | 0.01 | Linear | Poly-7 |

#### 4.1.1 TRAINING AND EVALUATION DATASETS

**Training.** To train our prior, we create a diverse cohort of 7383 high-quality, quality-controlled 1 mm isotropic MRI scans, comprising of 5279 T1-weighted (T1w), 1516 T2-weighted (T2w), and 588 FLAIR images from public datasets; ADNI Weiner et al. (2017), HCP Essen et al. (2012), Chinese HCP Yang et al. (2024), ADHD200 Brown et al. (2012), AIBL Fowler et al. (2021), CO-BRE Sidhu (2018) MCIC Gollub et al. (2013), ISBI2015 challenge and OASIS3 LaMontagne et al. (2018). All images are skull-stripped, bias-field corrected, and affinely registered to an atlas template. Detailed processing steps are available in Appendix A.2.

To showcase the versatility and robustness of our method, we perform experiments on a selection of challenging datasets. For each dataset, we have paired target and low-resolution images.

**Image restoration.** For image restoration tasks, we test our method on two datasets; a Clinical dataset and a Low-field dataset. The clinical dataset (ages 5-82) contains paired high- (1 mm

isotropic) and low-resolution scans with greater slice spacing and thickness, acquired with T1w (N=41), T2w (N=33), or FLAIR contrast (N=31). The low-resolution scans were acquired axially with voxel spacings provided in Appendix A.3. The Low-field dataset (ages 23-53, N=32 total scans) consists of paired low- and high-field T1w (N=16) and T2w (N=16) images acquired in healthy subjects. Low-field images were acquired at 0.064 T (Hyperfine Inc) either isotropically (3 mm) or axially (1.6, 1.6, 5 mm). High-field isotropic (1 mm) images were acquired at 3 T (Siemens Prisma), as described in previous work Sorby-Adams et al. (2024). Example figures as well as further details on data preprocessing and dataset descriptions, including demographics information, are available in Appendix A.3.

**Image inpainting and refinement.** We evaluate our inpainting approach on brain lesion datasets, where the goal is to reconstruct healthy tissue in regions affected by pathology. We conduct experiments on binary manual chronic strokes lesion segmentations and T1w images from the BraTS Baid et al. (2021) (N=398) and ATLAS Liew et al. (2018) (N=646) datasets. For the image refinement task, we first apply SynthSR to a subset of the ATLAS dataset and then apply our method with the forward model given in Equation 16 ($\tau_s$=0.05, set heuristically) to refine the images.

### 4.1.2 EVALUATION METRICS

To evaluate image restoration and refinement, we compare generated images from degraded scans with the original high-resolution 1 mm isotropic scans. We compute standard image quality metrics (IQMs): mean absolute error (MAE), peak signal-to-noise ratio (PSNR), structural similarity (SSIM) Wang et al. (2004), visual information fidelity (VIF) Sheikh & Bovik (2006), gradient magnitude similarity deviation (GMSD) Xue et al. (2014), and learned perceptual image patch similarity (LPIPS) Zhang et al. (2018) using an AlexNet backbone Krizhevsky et al. (2012). For metrics designed for 2D images, we adopt a 2.5D approach.

For inpainting, the goal is to not only inpaint disease regions but also produce anatomically plausible reconstructions. We generate pseudo-healthy images for each method and evaluate them using two unsupervised anomaly detection models—a VAE Baur et al. (2021) and an LDM Graham et al. (2023) with pretrained weights from Lawry Aguila et al. (2025). Successful inpainting should yield pseudo-healthy images within the natural variation of healthy anatomy, resulting in minimal detected anomalies by the anomaly detection methods. For each model we compute anomaly maps, we report MAE, LPIPS, and the maximum Dice for the respective method between the anomaly map and segmentation; here, lower Dice scores indicate effective removal of disease-related anomalies.

### 4.1.3 COMPARISON WITH STATE-OF-THE-ART METHODS

**Image restoration.** We compare our method to both data-driven and classical baselines designed for medical imaging. For classical approaches, we compare to UniRes Brudfors et al. (2019), a principled inverse problem solving approach to super resolution of clinical images which uses a total variation (TV) prior. In terms of data-driven methods, we compare to SynthSR, a data-driven machine learning method for joint SR and inpainting of heterogeneous T1w scans, two generative models which require paired images for training LoHiResGAN Islam et al. (2023) and Res-SRDiff Safari et al. (2025), a GAN and diffusion model approach respectively, and Di-Fusion Wu et al. (2025) a self-supervised denoising diffusion model approach trained on noisy data. For methods not designed for a specific modality, we exclude them from the corresponding analysis.

**Inpainting.** To assess our anomaly inpainting performance, we compare to SynthSR as well as two recently proposed diffusion model approaches; DDPM-2D Durrer et al. (2024a) and DDPM-pseudo3D Zhu et al. (2023). All baselines use paired images and segmentation maps during training.

## 4.2 IMAGE RESTORATION RESULTS

IQM values comparing generated to ground-truth high-resolution scans are shown in Table 2. Our method outperforms baselines across several metrics, achieving the highest, or joint highest, rank for all datasets. Data-driven methods, LoHiResGAN, Res-SRDiff and Di-Fusion, fail to generalise to these cohorts, as illustrated by their poor performance. SynthSR, although outperforming competitively on some IQMs (and outperforming our method in VIF for T1w Clinical), is restricted to predicting T1w intensities. UniRes is often a close performing baseline, which is expected given

that it also models image restoration explicitly with a forward model similar to ours. UniRes outperforms our method in some VIF values, whereas our method consistently achieves second best performance in this metric. For all other IQMs across all datasets, our method achieves the best performance, in some instances by quite considerable margins.

Qualitative T1w results are shown in Figure 2, with further examples for other modalities in Appendix A.8. LoHiResGAN and Res-SRDiff produce unrealistic images with severe artifacts, likely arising from bias fields, sharp intensity artifacts, and other noise not present during training. UniRes generates oversmoothed images, likely due to its TV prior and its reliance on information from multiple input modalities, whereas we apply it unimodally. Di-Fusion shows less pronounced but still notably blurry, voxelated reconstructions which lack the fine-grained details generate by our method. This is likely, in part, due to our use of synthetic rather than real noisy training data, which the method was designed for. As such data is scarce, and in our case unavailable, this requirement represents a significant limitation of Di-Fusion. SynthSR, whilst not as well as our method, does preserve key anatomical structures. However, our difference maps show reduced contrast, further supporting the strong quantitative results shown in Table 2.

### 4.3 IMAGE INPAINTING RESULTS

Inpainting results are given in Table 3. Our method achieves the best overall performance, attaining the highest rank on both datasets. For ATLAS, our method outperforms all baselines with im-

Table 2: Super-resolution results for the Clinical and Low-field MR datasets. For each modality and metric, green indicates the best results, and red indicates the second best performance.

| | Modality | Method | MAE ($\downarrow$) | PSNR ($\uparrow$) | SSIM ($\uparrow$) | LPIPS ($\downarrow$) | VIF ($\uparrow$) | GMSD ($\downarrow$) | Rank ($\uparrow$) |
|---|---|---|---|---|---|---|---|---|---|
| Clinical dataset | T1w | SynthSR | 0.1229 | 16.9876 | 0.1458 | 0.1834 | 0.1527 | 0.2660 | 3.00 |
| | | UniRes | 0.1948 | 11.1006 | 0.5101 | 0.4260 | 0.0930 | 0.3601 | 4.67 |
| | | LoHiResGAN | 0.0938 | 18.0808 | 0.1249 | 0.3984 | 0.0656 | 0.3536 | 4.00 |
| | | Res-SRDiff | 0.1825 | 13.1292 | 0.0608 | 0.6786 | 0.0477 | 0.3526 | 5.33 |
| | | Di-Fusion | 0.0849 | 17.0014 | 0.4231 | 0.2301 | 0.0907 | 0.2604 | 2.83 |
| | | Ours | 0.0450 | 20.9624 | 0.7501 | 0.1477 | 0.1177 | 0.2165 | 1.17 |
| | T2w | UniRes | 0.0355 | 20.8495 | 0.7509 | 0.2112 | 0.3238 | 0.3079 | 2.33 |
| | | LoHiResGAN | 0.1752 | 12.6202 | 0.0768 | 0.6601 | 0.0090 | 0.3841 | 5.00 |
| | | Res-SRDiff | 0.0803 | 19.1157 | 0.1439 | 0.4317 | 0.0979 | 0.3344 | 3.83 |
| | | Di-Fusion | 0.0475 | 21.1661 | 0.3914 | 0.1872 | 0.0788 | 0.2584 | 2.67 |
| | | Ours | 0.0252 | 23.7406 | 0.8073 | 0.1098 | 0.1355 | 0.1949 | 1.17 |
| | FLAIR | UniRes | 0.0951 | 15.7761 | 0.6624 | 0.2827 | 0.2992 | 0.3191 | 2.50 |
| | | Res-SRDiff | 0.1611 | 13.7190 | 0.0824 | 0.5745 | 0.0940 | 0.3357 | 4.00 |
| | | Di-Fusion | 0.0569 | 19.7739 | 0.4221 | 0.1922 | 0.1061 | 0.2517 | 2.33 |
| | | Ours | 0.0429 | 21.6849 | 0.8141 | 0.1063 | 0.1816 | 0.1936 | 1.17 |
| Low-field dataset | T1w | SynthSR | 0.1603 | 14.3101 | 0.0761 | 0.2315 | 0.0725 | 0.3212 | 4.33 |
| | | UniRes | 0.1138 | 12.8471 | 0.5709 | 0.3077 | 0.1489 | 0.3038 | 3.33 |
| | | LoHiResGAN | 0.0879 | 18.5893 | 0.1175 | 0.4366 | 0.0740 | 0.3536 | 3.83 |
| | | Res-SRDiff | 0.2147 | 11.6563 | 0.0540 | 0.7800 | 0.0146 | 0.3736 | 6.00 |
| | | Di-Fusion | 0.0781 | 16.4754 | 0.4698 | 0.2027 | 0.1024 | 0.2295 | 2.50 |
| | | Ours | 0.0292 | 23.5889 | 0.8455 | 0.1042 | 0.1709 | 0.1855 | 1.00 |
| | T2w | UniRes | 0.0403 | 22.2391 | 0.6743 | 0.2215 | 0.1834 | 0.2912 | 2.50 |
| | | LoHiResGAN | 0.0826 | 19.1590 | 0.1249 | 0.3763 | 0.0708 | 0.3543 | 4.33 |
| | | Res-SRDiff | 0.0955 | 18.4865 | 0.1116 | 0.3763 | 0.1116 | 0.3172 | 4.50 |
| | | Di-Fusion | 0.0376 | 22.2704 | 0.4382 | 0.1854 | 0.0823 | 0.2224 | 2.50 |
| | | Ours | 0.0276 | 23.2194 | 0.8544 | 0.1194 | 0.1422 | 0.1592 | 1.17 |

Table 3: Inpainting results for inpainting of the BraTS and ATLAS datasets.

| Dataset | Method | VAE$_{MAE}$ ($\downarrow$) | VAE$_{LPIPS}$ ($\downarrow$) | VAE$_{Dice}$ ($\downarrow$) | LDM$_{MAE}$ ($\downarrow$) | LDM$_{LPIPS}$ ($\downarrow$) | LDM$_{Dice}$ ($\downarrow$) | Rank ($\downarrow$) |
|---|---|---|---|---|---|---|---|---|
| BraTS | SynthSR | 0.0947 | 0.3692 | 0.1466 | 0.1071 | 0.2234 | 0.0310 | 2.33 |
| | DDPM-2D | 0.1122 | 0.3723 | 0.1707 | 0.1042 | 0.2291 | 0.1862 | 3.50 |
| | DDPM-3D | 0.1025 | 0.3615 | 0.1736 | 0.0932 | 0.2127 | 0.1913 | 2.83 |
| | Ours | 0.0705 | 0.3428 | 0.1579 | 0.0677 | 0.1804 | 0.0834 | 1.33 |
| ATLAS | SynthSR | 0.1049 | 0.4047 | 0.0486 | 0.1221 | 0.2309 | 0.0037 | 3.17 |
| | DDPM-2D | 0.1162 | 0.3956 | 0.0491 | 0.1078 | 0.2283 | 0.0448 | 3.17 |
| | DDPM-3D | 0.1012 | 0.3808 | 0.0493 | 0.0900 | 0.2046 | 0.0470 | 2.67 |
| | Ours | 0.0615 | 0.3492 | 0.0473 | 0.0502 | 0.1657 | 0.0018 | 1.00 |

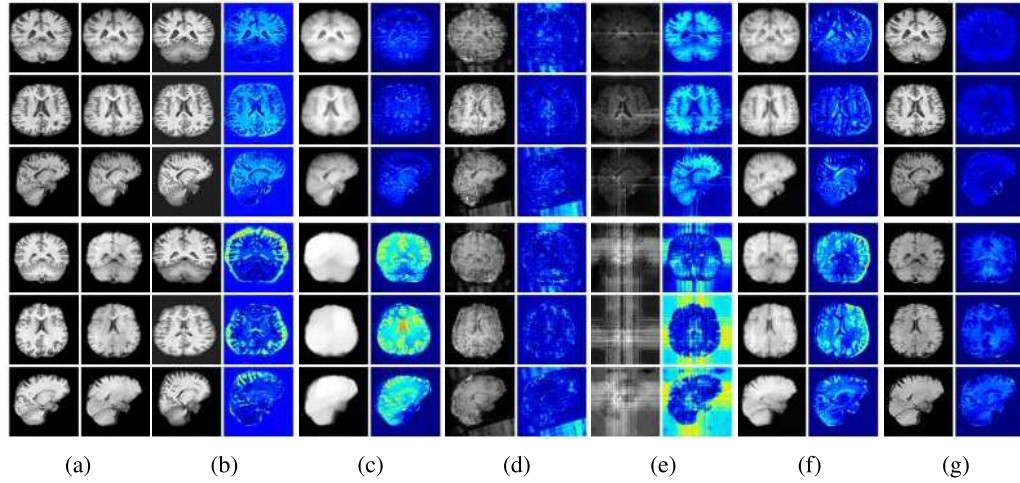

(a)    (b)    (c)    (d)    (e)    (f)    (g)

Figure 2: Example restoration results for the clinical (top) and Low-Field MR dataset (bottom). Each column shows a stacked pair of images (top/bottom) corresponding to a different method. (a) Ground truth T1w (1mm) image and linearly interpolated low-resolution image, (b) SynthSR, (c) UniRes, (d) LoHiResGAN, (e) Res-SRDiff, (f) Di-Fusion, and (g) Ours. Difference maps are shown for each method.

provements of 39.2% ($\text{VAE}_{\text{MAE}}$), 8.3% ($\text{VAE}_{\text{LPIPS}}$), 2.7% ($\text{VAE}_{\text{Dice}}$), 44.2% ($\text{LDM}_{\text{MAE}}$), 19.0% ($\text{LDM}_{\text{LPIPS}}$), and 51.4% ($\text{LDM}_{\text{Dice}}$). On BraTS, it improves over the best baselines by 25.6% ($\text{VAE}_{\text{MAE}}$), 5.2% ($\text{VAE}_{\text{LPIPS}}$), 27.4% ($\text{LDM}_{\text{MAE}}$), and 15.2% ($\text{LDM}_{\text{LPIPS}}$), while remaining competitive on the remaining metrics.

Figure 3 (additional examples in Appendix A.9) shows that SynthSR preserves healthy tissue but struggles with large lesions, while DDPM-2D and DDPM-3D, despite producing high-contrast anomaly maps, generate unrealistic homogeneous inpainting, consistent with their lower performance in Table 3. In contrast, our method yields the most anatomically plausible inpainted regions, although anomaly maps appear subtle due to low contrast between lesions and healthy tissue.

### 4.4 IMAGE REFINEMENT RESULTS

The image refinement results (see Appendix A.10 for more examples) in Figure 4 highlight our framework's ability to enhance outputs from existing methods. While SynthSR can inpaint disease regions, the resulting tissue often appears unrealistic. Our method further refines these areas, producing anatomically plausible reconstructions with more realistic surface structures.

### 4.5 HYPERPARAMETER STUDIES

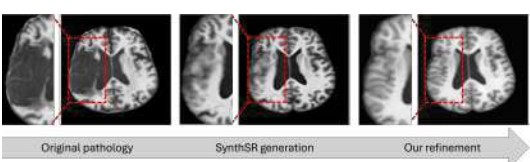

Figure 4: Example ATLAS refinement result.

Figure 5 reports results across a range of likelihood precision values $\tau$, informed by prior work Zheng et al. (2025) and synthetic data (see Appendix A.4). Although restoration tasks tend to perform best at $\tau = 0.01$, both restoration and inpainting achieve strong results at $\tau = 0.005$, providing a good balance between data fidelity and prior regularization. We therefore use $\tau = 0.005$ in our analysis to ensure that our method can be applied across these tasks without requiring hyperparameter tuning.

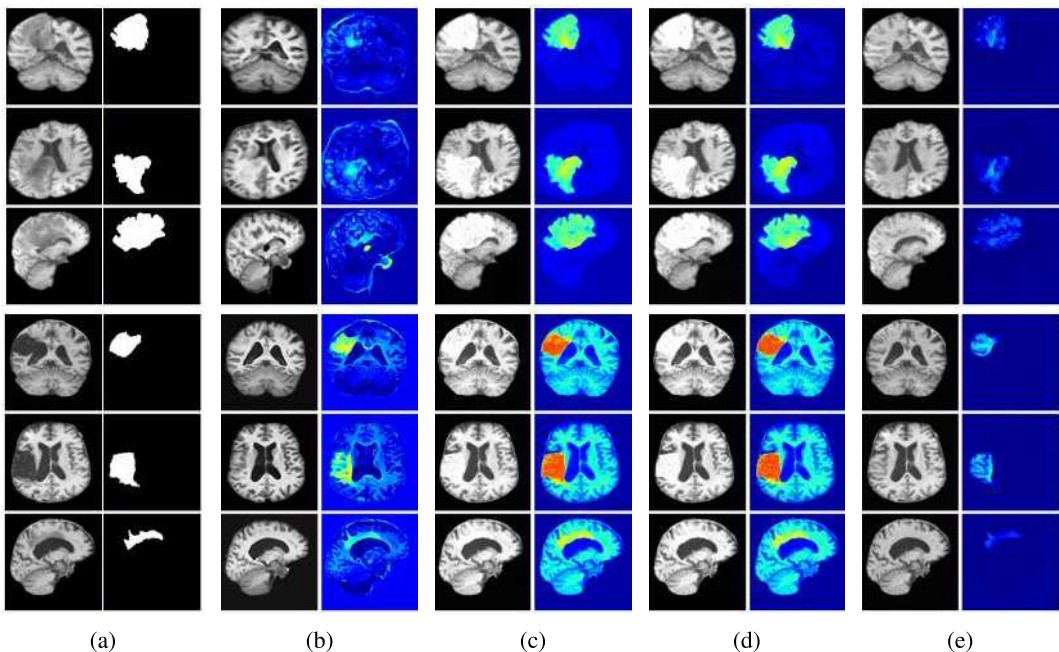

(a)       (b)       (c)       (d)       (e)

Figure 3: Example inpainting results for the BraTS (top) and ATLAS (bottom) datasets. (a) Original image and manual segmentation map, (b) SynthSR, (c) DDPM-2D, (d) DDPM-3D and (e) Ours. Reconstructions and difference maps are shown for each method.

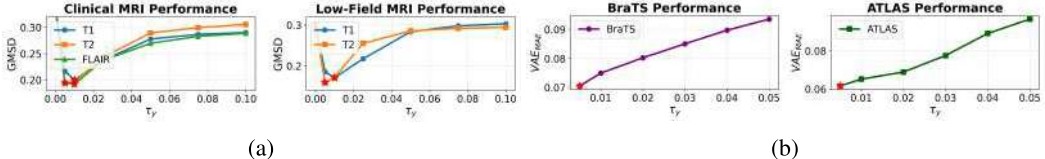

(a)                                           (b)

Figure 5: $\tau_y$ performance for (a) restoration and (b) inpainting tasks.

## 5 CONCLUSION

We present the first general-purpose application of diffusion models as priors for medical imaging inverse problems in neuroimaging. Our approach integrates powerful data-driven priors learned from diverse brain MRI with flexible forward models to tackle a range of imaging challenges. Importantly, our method does not require acquisition parameters or paired training data and can be applied directly to degraded scans. Extensive experiments on heterogeneous, noisy datasets demonstrate that our proposed method achieves state-of-the-art performance compared to competitive baseline methods. Limitations and further work is discussed in Appendix A.7. By flexibly improving low-resolution or otherwise suboptimal scans, our method has the potential to significantly advance both clinical practice and research, for example by reducing scan times, enabling retrospective analysis of archived datasets, or supporting studies in populations where high-quality imaging is difficult to obtain.

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

# A APPENDIX

**LLM usage.** In this work we use LLMs for text refinement and generation, coding, and problem solving.

**Code.** We base our code on the InverseBench (`https://github.com/devzhk/InverseBench`) and UniRes packages (`https://github.com/brudfors/UniRes`). It is available at: `https://anonymous.4open.science/r/iclr2026-E1BC`.

## A.1 TRAINING AND SAMPLING DETAILS

Training of the diffusion prior was performed on a single NVIDIA A100 GPU (80 GB) with a batch size of 1 using the hyperparameters described in Section 4.1. Inference for all methods was conducted on NVIDIA Quadro RTX 8000 GPUs (48 GB). Model sizes for our method and all baselines are summarized in Table 4. Per-sample inference times for our method and all baselines are reported in Table 5.

Table 4: Model sizes for our diffusion prior and baseline methods.

| Task | Model | Parameters (M) | Size (MB, FP32) |
|---|---|---|---|
| - | Diffusion prior (Ours) | 52.25 | 199 |
| - | SynthSR | 26.48 | 101 |
| Restoration | UniRes | - | - |
| | LoHiResGAN | 54.42 | 207 |
| | Res-SRDiff | 347.21 | 1324 |
| | Di-Fusion | 46.17 | 176 |
| Inpainting | DDPM-2D | 113.67 | 433 |
| | DDPM-3D | 137.48 | 524 |

Table 5: Per-sample inference times for posterior sampling method and baseline methods.

| Task | Model | Time per sample (s) |
|---|---|---|
| Restoration | Ours | 1193.4 |
| | SynthSR | 18.0 |
| | UniRes | 174.0 |
| | LoHiResGAN | 17.3 |
| | Res-SRDiff | 300.2 |
| | Di-Fusion | 3.8 |
| Inpainting | Ours | 989.2 |
| | SynthSR | 18.0 |
| | DDPM-2D | 1983.2 |
| | DDPM-3D | 4924.9 |

## A.2 TRAINING DATASETS

The number of scans from each dataset are provided in Table 6. Each image is skull-stripped and bias-field corrected with FreeSurfer Fischl (2012) and N4ITK Tustison et al. (2010) respectively, and min-max normalized to [-1,1], All volumes are affinely registered the MNI305 template Evans et al. (1993) using EasyReg Iglesias (2023) and transformed and cropped to $176^3$ voxels. The affine transformation to MNI305 space is recomputed by aligning the centroids of anatomical labels from SynthSeg Billot et al. (2023) segmentations to the corresponding atlas centroids.

Table 6: Summary of MRI scans in the training data by dataset and modality

| Dataset | T1w | T2w | FLAIR |
|---|---|---|---|
| ABIDE | 819 | – | – |
| AIBL | 820 | – | – |
| HCP | 1033 | 821 | – |
| OASIS3 | 1238 | 695 | 273 |
| ADNI3 | 316 | – | 315 |
| Buckner40 | 38 | – | – |
| Chinese-HCP | 212 | – | – |
| ISBI2015[a] | 21 | – | – |
| MCIC | 161 | – | – |
| **Total** | **5279** | **1516** | **588** |

[a]https://biomedicalimaging.org/2015/program/isbi-challenges/

### A.3    DATA FOR POSTERIOR SAMPLING

The experiments in this work use four datasets: two in-house datasets for image restoration (a Clinical cohort and a Low-field cohort), and two open-source datasets for inpainting and refinement (BraTS and ATLAS). In this section we provide additional information on these datasets.

For both the Clinical and Low-field datasets, low-resolution images are skull-stripped and normalized to [-1, 1]. The alignment to MNI space is required by forward model given in Equation 11 and is achieved by recomputing the affine transformation through centroid alignment of anatomical labels from SynthSeg Billot et al. (2023) segmentations with the corresponding atlas centroids. Example low-resolution images are shown in Figure 6.

At inference, super-resolved degraded scans are affinely registered to MNI space if not already aligned. In cases where super-resolved images were too poor in quality for direct registration, we instead applied the inverse affine transform obtained by registering the high-resolution image to its low-resolution counterpart using NiftyReg Ourselin et al. (2001; 2002).

Table 7: Demographics of Clinical and Low-field MRI Datasets

| Dataset | Modality | Age Range (yrs) | Racial Split (White / Black / Asian / Other) | Total # Scans |
|---|---|---|---|---|
| Clinical | T1 | 5 − 82 | 19 / 15 / 7 / 0 | 41 |
| Clinical | T2 | 21 − 63 | 15 / 10 / 8 / 0 | 33 |
| Clinical | FLAIR | 3 − 73 | 15 / 5 / 11 / 0 | 31 |
| Low-field | T1/T2/FLAIR | 23 − 53 | 24 / 0 / 6 / 2 | 32 |

Images from the ATLAS and BraTS datasets are skull-stripped, bias field corrected, and affinely registered to MNI space. For the BraTS dataset, scans were manually QCed for limited noise and artifacts.

### A.4    SYNTHETIC DATA FOR BASELINE TRAINING AND HYPERPARAMETER ANALYSIS

Synthetic low-resolution MRI data are generated from the high-resolution scans describe in A.2 using frequency-domain filtering and spatial downsampling to simulate thick-slice acquisition.

First, the image is transformed to Fourier space using a 3D FFT with frequency shifting. A 3D Gaussian low-pass filter is then applied to approximate the point-spread function and slice profile, attenuating high-frequency components that would otherwise cause aliasing during resampling. The filtered frequency representation is then returned to the spatial domain via inverse frequency shift and inverse FFT.

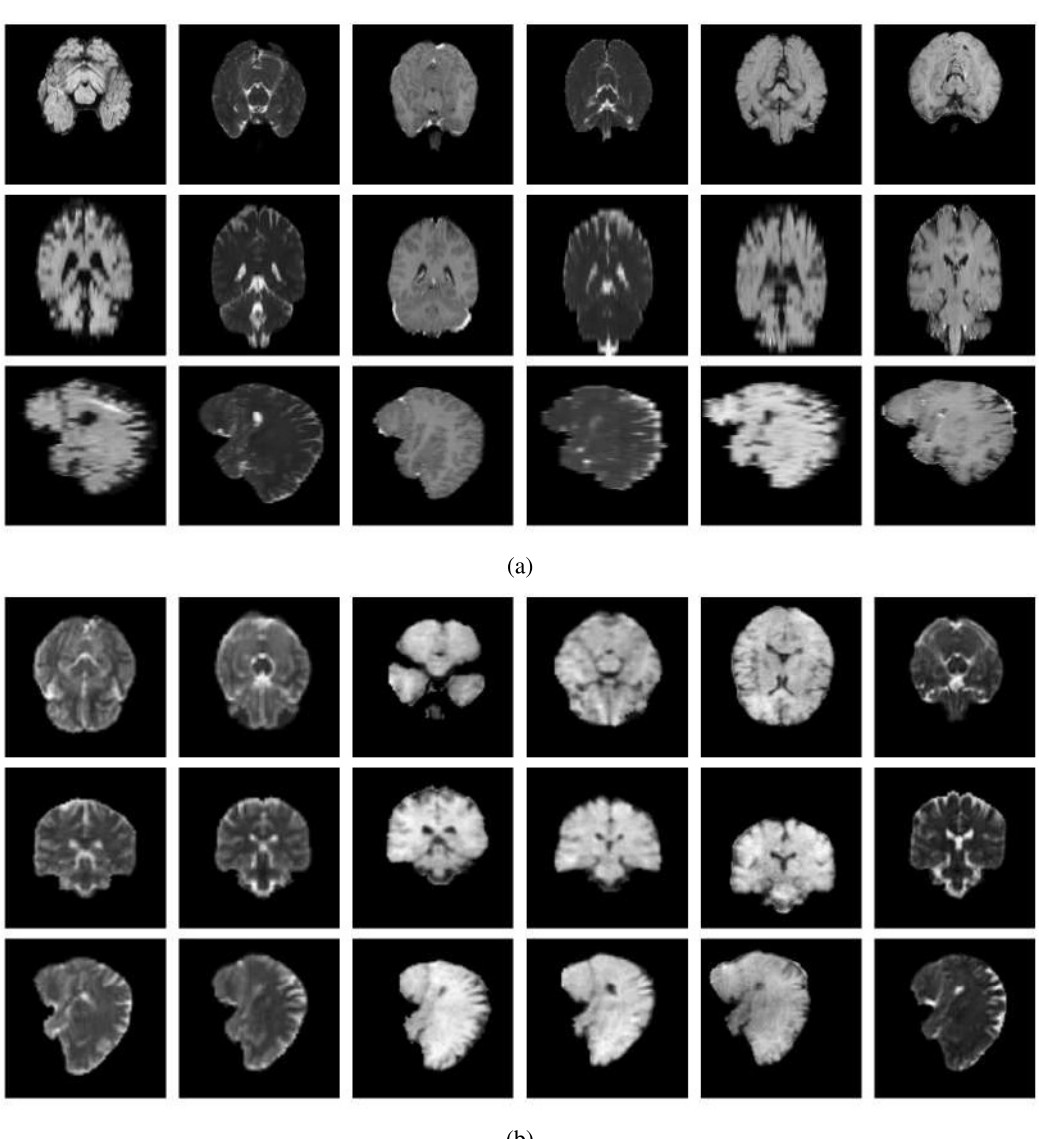

(a)

(b)

Figure 6: Example low-resolution images from the (a) Clinical and (b) Low-field datasets. Both cohorts exhibit clear registration requirements, downsampling, low signal, and bias-field artifacts, highlighting the challenges of image restoration in such heterogeneous and noisy data.

Table 8: Voxel spacing and number of scans per dataset and modality

| Dataset | Modality | Voxel Spacing (mm) | # Scans |
|---------|----------|--------------------|---------|
| Clinical | T1 | (1.375, 1.375, 6.0) | 35 |
| Clinical | T1 | (1.0, 1.0, 3.0) | 6 |
| Clinical | T2 | (0.977, 0.977, 3.0) | 20 |
| Clinical | T2 | (1.429, 1.429, 5.0) | 10 |
| Clinical | T2 | (1.6, 1.6, 6.0) | 3 |
| Clinical | FLAIR | (1.375, 1.375, 6.0) | 18 |
| Clinical | FLAIR | (1.0, 1.0, 3.0) | 12 |
| Clinical | FLAIR | (1.6, 1.6, 6.0) | 1 |
| Low-field | T1/T2/FLAIR | (2.0, 2.0, 2.0) | 30 |
| Low-field | T1/T2/FLAIR | (1.6, 1.6, 5.0) | 2 |

Next, the image is spatially downsampled with trilinear interpolation to match the target resolution. Output dimensions are set as ⌊original size/factor⌋ along each axis, where the factors correspond to the ratio of original to target voxel spacing.

For the hyperparameter analysis, we simulate axially acquired samples of voxel spacing $(1.6, 5.0, 1.6)$ mm. For the baseline training, we samples factors stochastically from realistic ranges.

Finally, we apply a smooth multiplicative bias field, simulated by sampling random 3rd order polynomial coefficients, to model intensity inhomogeneities commonly observed in MRI acquisitions.

### A.5  Baseline Methods

**SynthSR.** We use the implementation available with FreeSurfer 7.4.1. Since SynthSR generates the skull, we use SynthSeg Billot et al. (2023) for skull stripping of all generated images to ensure consistent preprocessing with other methods.

**UniRes.** We use the original implementation available at `https://github.com/brudfors/UniRes`. For fair comparison with our approach and other baselines, we use a uni-modal configuration with default hyperparameters settings from the GitHub repository.

**LoHiResGAN.** We use the original codebase and pre-trained model weights from `https://github.com/khtohidulislam/LoHiResGAN`. All input samples are registered to the reference test image provided with the original implementation.

**Res-SRDiff.** We use the original codebase available at `https://github.com/mosaf/res-srdiff`. Since pre-trained weights were not publicly available, we train the model from scratch using high-resolution and synthetic low-resolution image pairs described in Section A.2 and A.4. All training data is pre-registered to MNI space.

**Di-Fusion.** We use the original codebase available at `https://github.com/FouierL/Di-Fusion`. Similarly to Res-SRDiff, train a model from scratch using slices of synthetic low-resolution images described in Section A.4, pre-registering all data to MNI space.

**DDPM-2D and DDPM-Pseudo3D.** We use the implementation, inference code, and pre-trained network weights from Durrer et al. (2024b) without modification.

### A.6  Ablation of bias field effects

We conduct an ablation study of Equation 11 with and without the bias field, **b**, for the T1w clinical cohort. Table 9 shows that for the majority of the IQM, we achieve improved performance when including bias field effects in our likelihood formulation.

Table 9: Ablation of modelling bias field effects.

| Method | MAE (↓) | LPIPS (↓) | SSIM (↑) | PSNR (↑) | VIF (↑) | GMSD (↓) |
|---|---|---|---|---|---|---|
| Bias field | 0.0450 | 0.1477 | 0.7501 | 20.9624 | 0.1177 | 0.2165 |
| No Bias field | 0.0590 | 0.1502 | 0.7618 | 19.1060 | 0.1227 | 0.2205 |

### A.7  Further discussion

There are number of limitations and directions of further work which warrant discussion. Firstly, our methods ability to generate realistic tissue contrasts requires improvement. We plan to investigate more sophisticated likelihood formulations that better preserve contrast characteristics, and improved training of the prior to capture a wider range of potential image contrasts. Additionally, sampling time remains slow due to the iterative nature of diffusion-based posterior sampling, which may limit real-time clinical applications. Future work will focus on exploring consistency models Song et al. (2023) to accelerate sampling and develop adaptive hyperparameter selection strategies. Additionally, we will conduct downstream analyses of the generated images to further evaluate their clinical utility, including assessment of how well enhanced images perform in standard neuroimaging pipelines such as image segmentation.

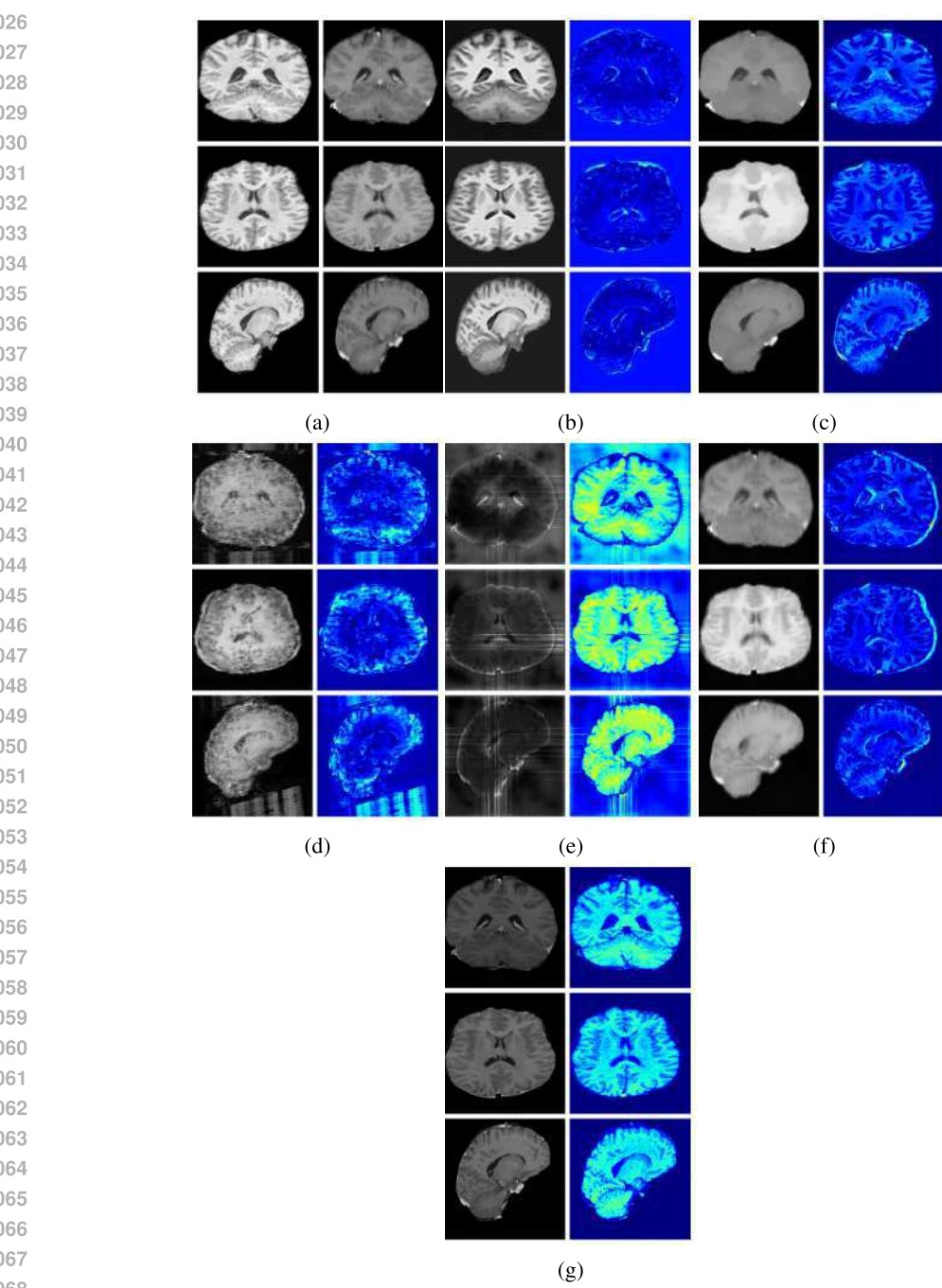

Figure 7: Example restoration results for T1w images from the Clinical dataset. (a) Original T1w (1mm) image and linearly interpolated low-resolution image, (b) SynthSR, (c) UniRes, (d) LoHiRes-GAN, (e) Res-SRDiff, (f) Di-Fusion, and (g) Ours. Difference maps are shown for each method.

## A.8    ADDITIONAL QUALITATIVE RESTORATION RESULTS

Additional qualitative results for the Clinical dataset are given in Figures 7, 8 and 9, and for the Low-field dataset in Figures 10 and 11.

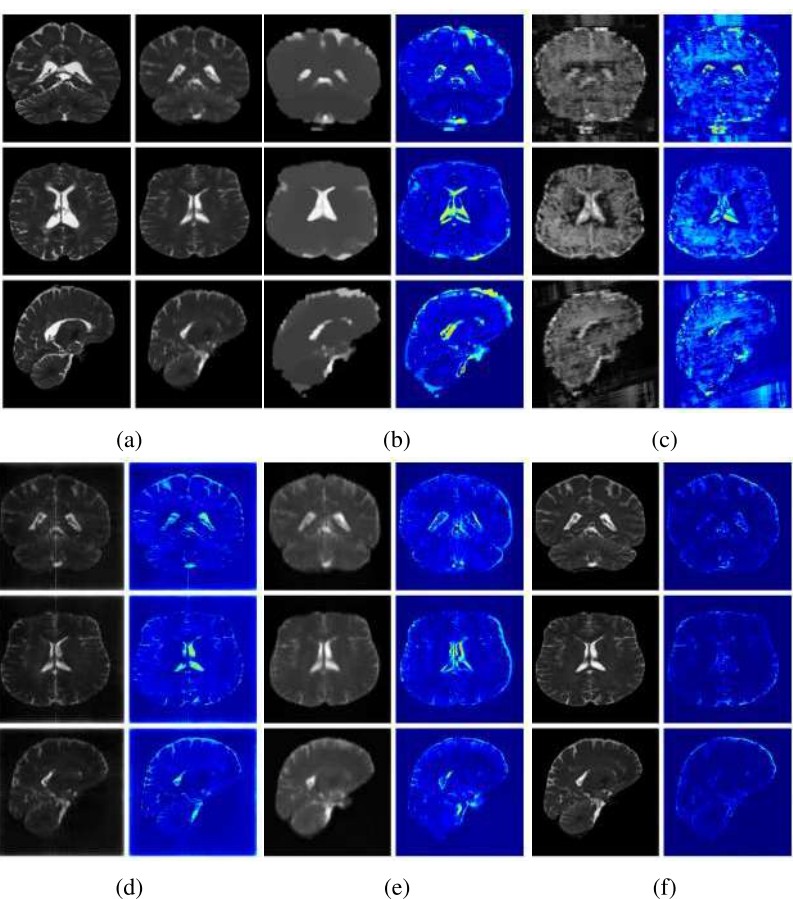

Figure 8: Example restoration results for T2w images from the Clinical dataset. (a) Original T2w (1mm) image and linearly interpolated low-resolution image, (b) UniRes, (c) LoHiResGAN, (d) Res-SRDiff, (e) Di-Fusion and (f) Ours. Difference maps are shown for each method.

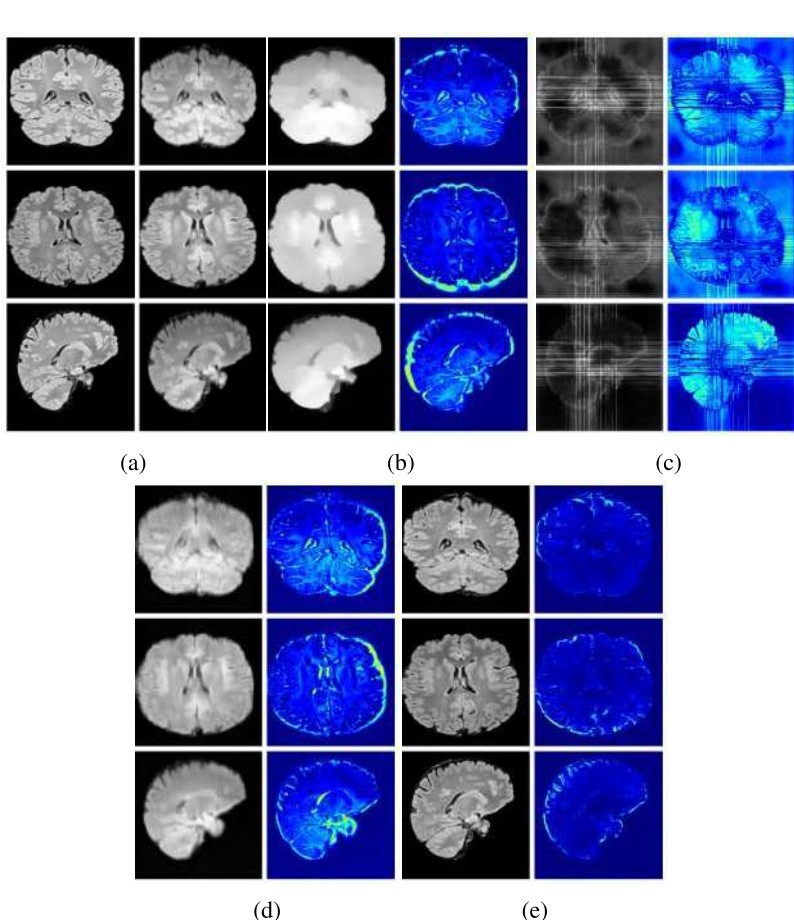

(a)                                  (b)                                  (c)

(d)                                  (e)

Figure 9: Example restoration results for FLAIR images from the Clinical dataset. (a) Original FLAIR (1mm) image and linearly interpolated low-resolution image, (b) UniRes, (c) Res-SRDiff, (d) Di-Fusion, and (e) Ours. Difference maps are shown for each method.

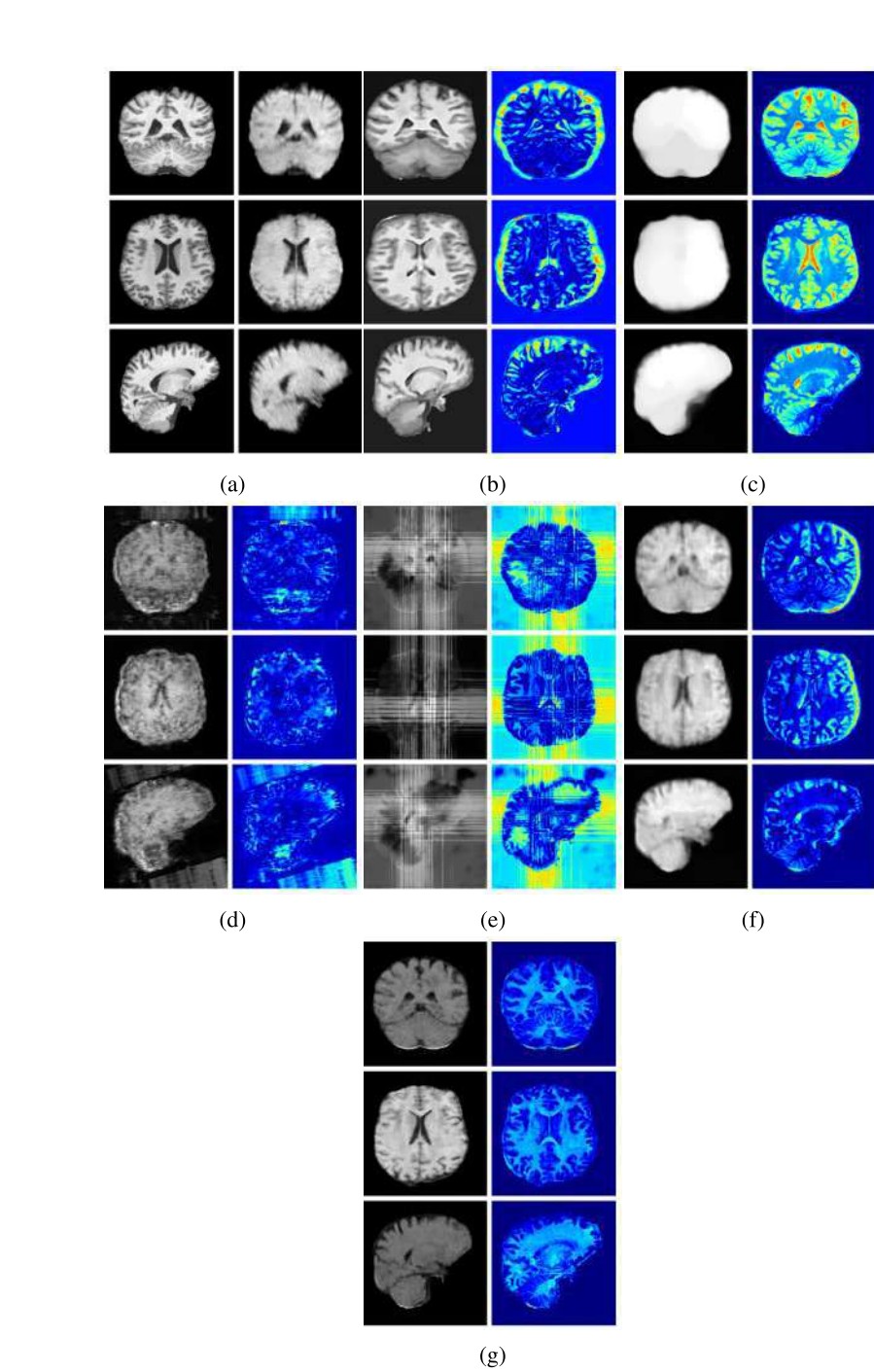

Figure 10: Example restoration results for T1w images from the Low-field dataset. (a) Original T1w (1mm) image and linearly interpolated low-resolution image, (b) SynthSR, (c) UniRes, (d) LoHiResGAN, (e) Res-SRDiff, (f) Di-Fusion, and (g) Ours. Difference maps are shown for each method.

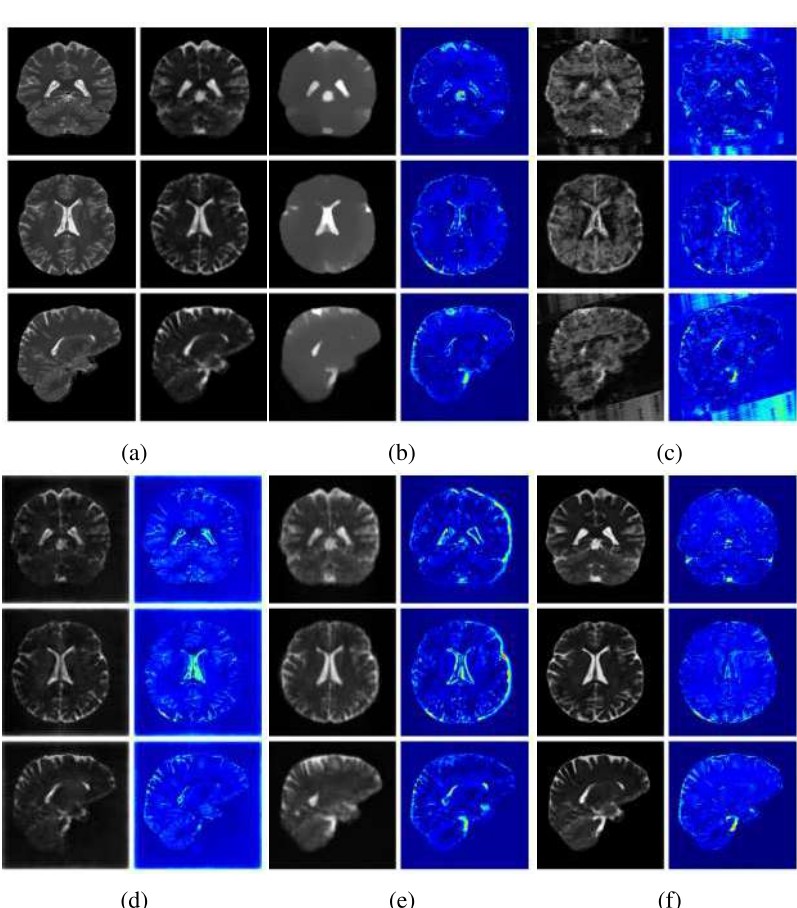

Figure 11: Example restoration results for T2w images from the Low-field dataset. (a) Original T2w (1mm) image and linearly interpolated low-resolution image, (b) UniRes, (c) LoHiResGAN, (d) Res-SRDiff, (e) Di-Fusion, and (f) Ours. Difference maps are shown for each method.

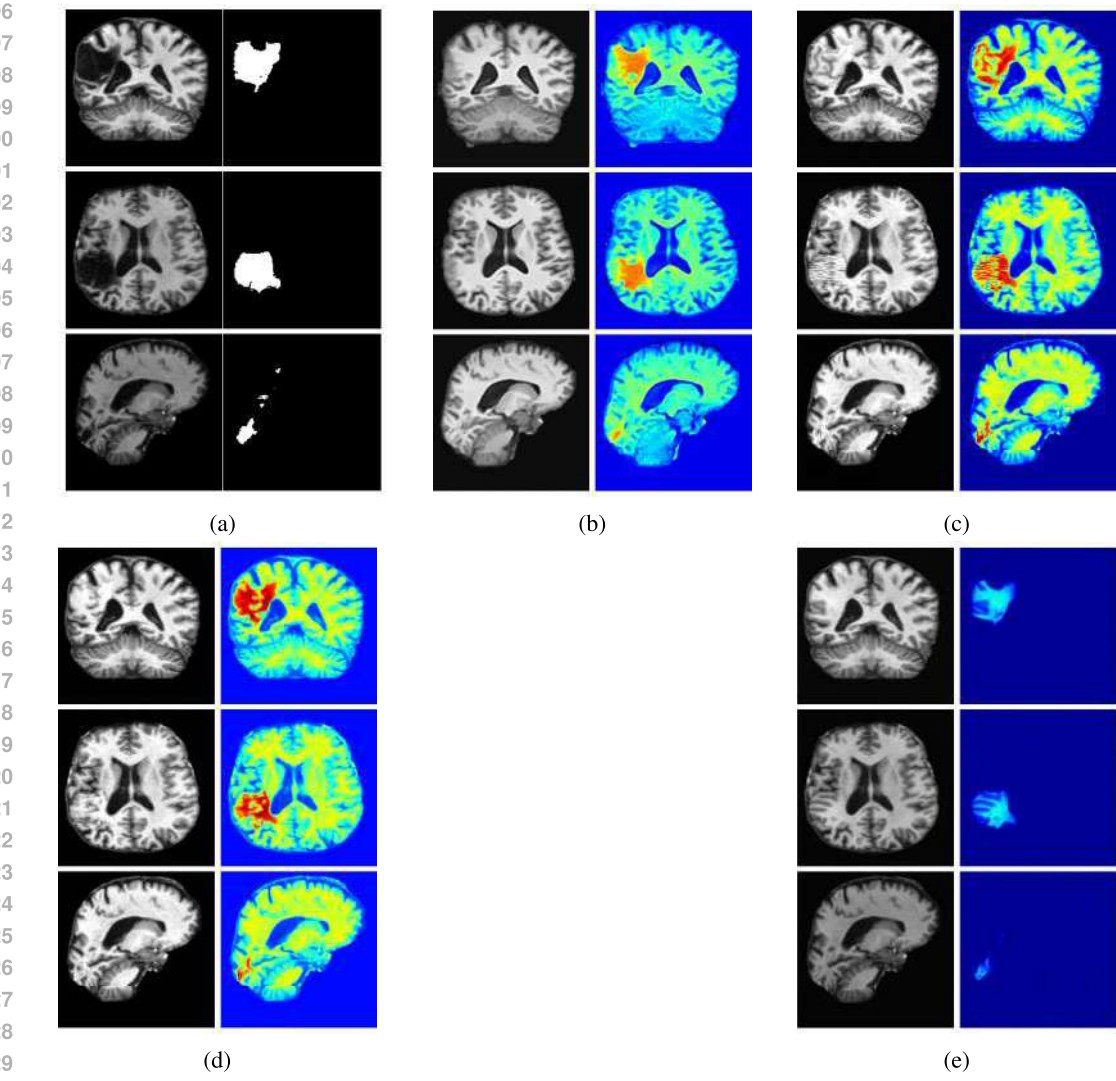

Figure 12: Example inpainting results for the ATLAS datasets. (a) Original image and manual segmentation map, (b) SynthSR, (c) DDPM-2D, (d) DDPM-3D and (e) Ours. Reconstructions and difference maps are shown for each method.

### A.9 ADDITIONAL QUALITATIVE INPAINTING RESULTS

Additional qualitative results for the ATLAS and BraTS datasets are given in Figures 12 and 13, respectively.

### A.10 ADDITIONAL QUALITATIVE REFINEMENT RESULTS

Additional qualitative refinement results for subjects from the ATLAS dataset are given in Figure 14

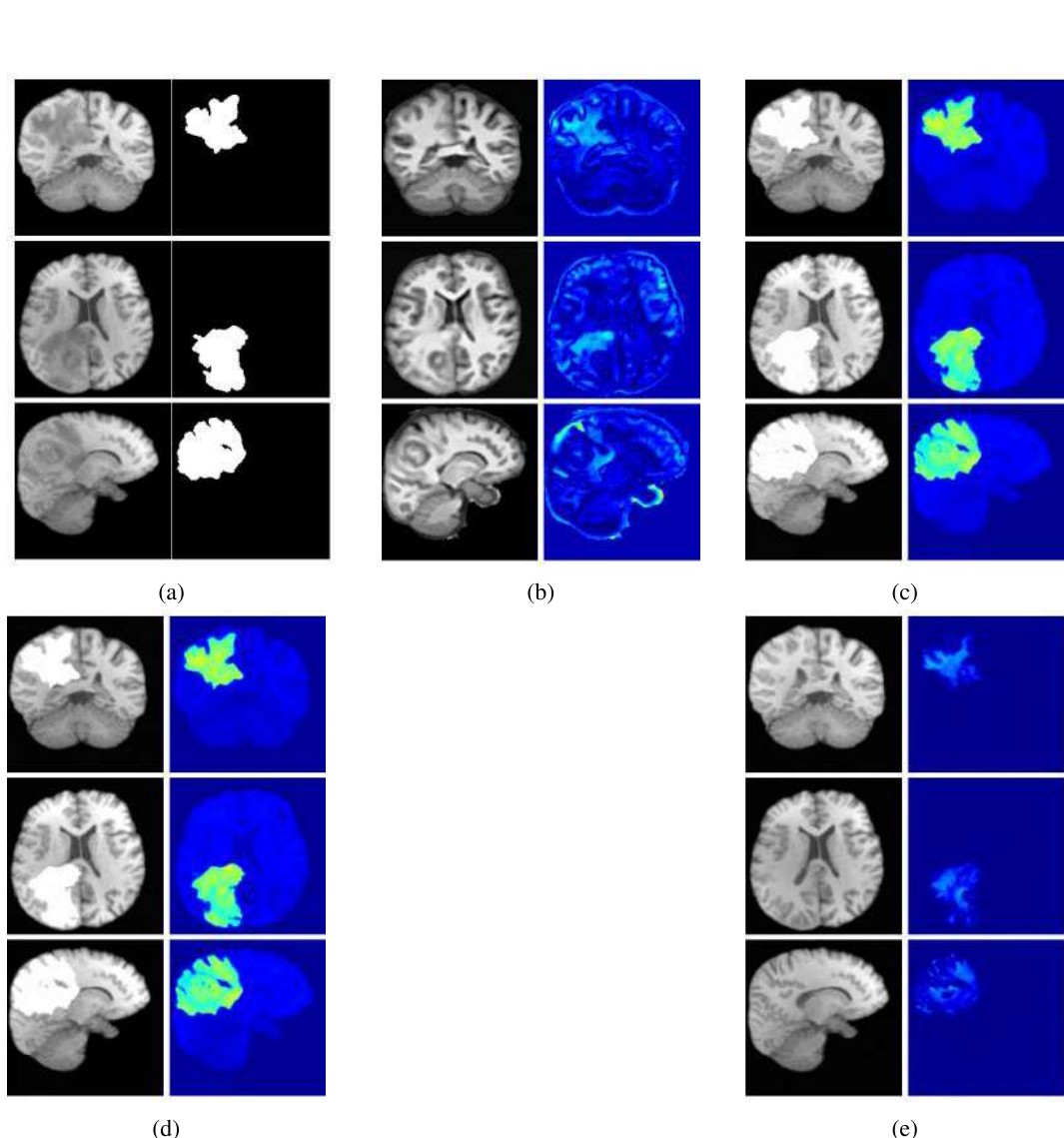

Figure 13: Example inpainting results for the BraTS datasets. (a) Original image and manual segmentation map, (b) SynthSR, (c) DDPM-2D, (d) DDPM-3D and (e) Ours. Reconstructions and difference maps are shown for each method.

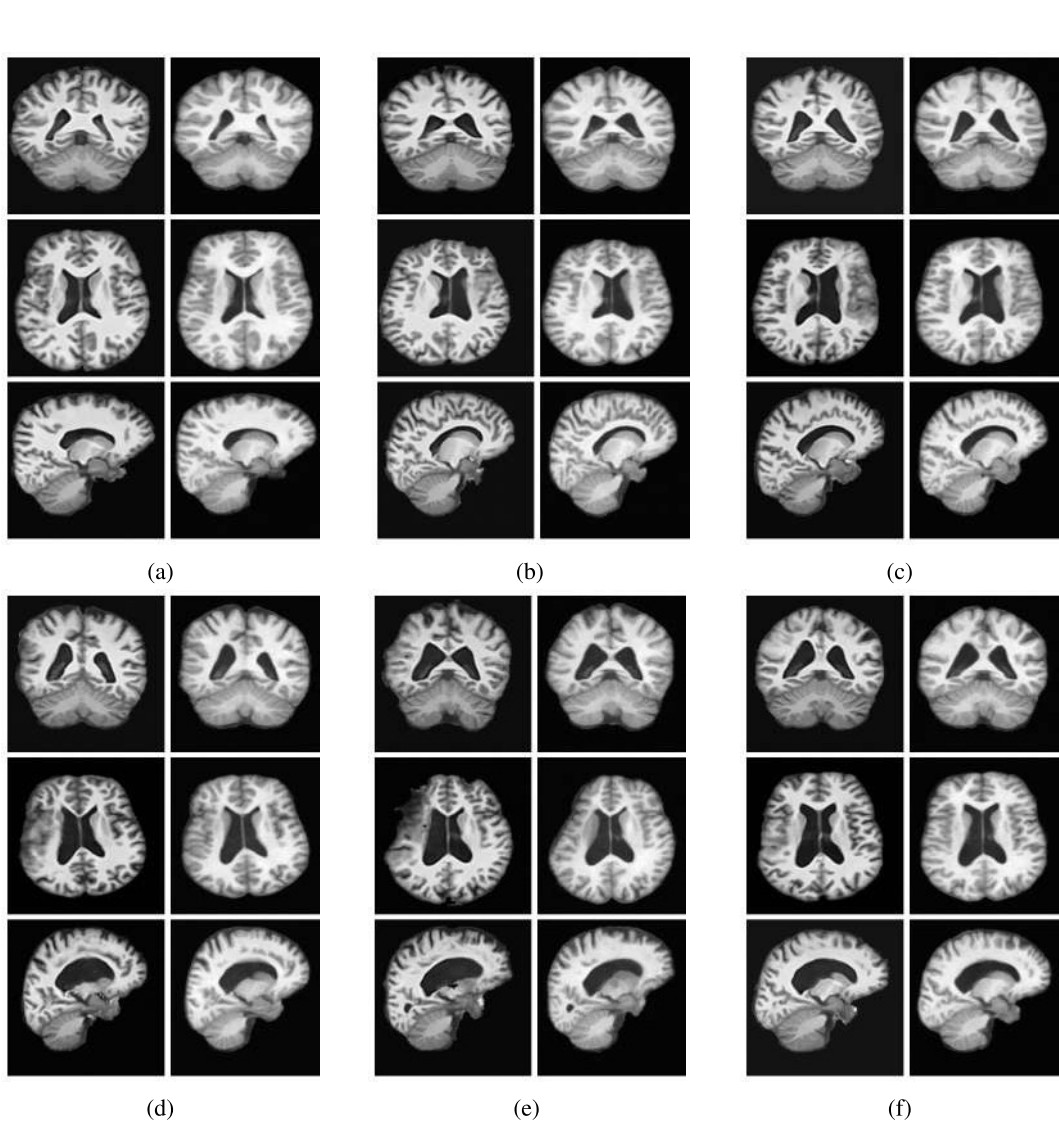

(a) (b) (c)

(d) (e) (f)

Figure 14: Example refinement results for subjects from the ATLAS dataset. For each subject, initial approximations generated by SynthSR are given in the left column and refined images generated by our method are given in the right column.

