# OpenReview forum: "Deep generative priors for 3D brain analysis"
_ICLR.cc/2026/Conference — Submitted to ICLR 2026_

### Official Review · Reviewer_DDwk · 2025-10-26

**Soundness:** 2
**Presentation:** 4
**Contribution:** 2
**Rating:** 2
**Confidence:** 4

**Summary:**

This paper explores the use of diffusion models as generative priors for solving 3D brain MRI inverse problems, including super-resolution, inpainting, and image refinement. The method combines a pre-trained 3D score-based diffusion prior with explicit forward models to incorporate domain knowledge, enabling the reconstruction of high-quality brain images without paired training data or acquisition parameters. The framework is validated across multiple heterogeneous datasets (clinical, low-field, and pathological MRI), showing improved quantitative metrics and enhanced anatomical fidelity compared to both classical and data-driven baselines.

**Strengths:**

- The experiments are extensive, covering various datasets and tasks with consistent improvements in image quality and anatomical plausibility over strong baselines.

**Weaknesses:**

- The main weakness of this work lies in its lack of clear technical novelty. The use of diffusion models as priors for inverse problems has already become a well-established paradigm, with several prior works applying similar formulations to medical and general imaging tasks. In this paper, the authors simply train a standard 3D diffusion model and apply existing posterior sampling methods to a few common MRI problems such as super-resolution and inpainting. There is no apparent methodological innovation or theoretical advancement beyond adapting known techniques to a different dataset. If this work were presented as a benchmark or large-scale evaluation study, it might be reasonable, but for a general ICLR track, the contribution appears incremental and lacks sufficient originality.

- In addition, the likelihood modeling and posterior sampling components rely on heuristic parameter tuning, without principled justification or adaptive estimation. This makes the framework sensitive to task-specific configurations and limits its reproducibility and general applicability. A deeper analysis or ablation on how the likelihood formulation impacts the final performance would be needed.

**Questions:**

Please see the weakness section.

---

> ### Author Response · Authors · 2025-11-20
> **Response to reviewer DDwk**
>
> We would like to thank the reviewer for their feedback. We are encouraged to hear that the reviewer thought our experiments were extensive with consistent improvements over strong baselines. We will respond to specific comments below.
>
> **[comment]** *“The use of diffusion models as priors for inverse problems has already become a well-established paradigm, with several prior works applying similar formulations to medical and general imaging tasks.”*
>
> **[response]** While we acknowledge that diffusion models have been applied as priors for inverse problems in imaging, we respectfully note that our literature review did not identify prior works addressing general MRI enhancement via posterior sampling with diffusion model priors and principled likelihoods (specifically designed to handle diverse scenarios) without requiring paired training data or access to acquisition parameters. Could the reviewer kindly share the works they are referring to?
>
> **[comment]** *"In this paper, the authors simply train a standard 3D diffusion model and apply existing posterior sampling methods to a few common MRI problems such as super-resolution and inpainting. There is no apparent methodological innovation or theoretical advancement beyond adapting known techniques to a different dataset.”*
>
> **[response]** We respectfully disagree with this characterization of our work. Our contribution is not 'simply applying' existing methods, but rather designing a unified posterior sampling framework with novel likelihood formulations specifically tailored to MRI inverse problems. Our key methodological innovations include:
> - **Principled forward models for image restoration inspired by MRI physics:** We introduce a likelihood (Eq. 11) where bias field parameters are jointly optimized during posterior sampling to simultaneously correct artifacts and enhance tissue contrast. We integrate this with the rotation-blurring-downsampling forward model (Eq. 9) to create a likelihood that jointly addresses both geometric degradation and intensity corruption. Our alternating updates for the image and bias-field coefficients follow a standard descent procedure, reliably lowering the joint objective during optimization. This provides a principled and stable way to refine both geometry and intensity together, rather than treating bias correction as a separate preprocessing step. Our combined formulation is particularly impactful for low-field MRI and certain clinical MRI sequences where poor tissue contrast hinders interpretation of scans.  This has not been addressed in existing methods.
> - **Constrained inpainting:** We introduce a constrained inpainting likelihood (Eq. 14-15) which enables anatomically-plausible disease region inpainting while preserving individual tissue characteristics. This is a task other state-of-the-art diffusion-based approaches have struggled with.
> - **Refinement of existing reconstructions:** We introduce a method (Eq. 16) to improve outputs from any existing algorithm by treating them as initial approximations. To our knowledge, our method is the first general framework for this.
>
> Beyond the likelihood formulations, our work shows that a single diffusion prior outperforms task-specific baselines across multiple inverse problems, modalities, and datasets. Releasing our trained prior alongside these likelihood formulations provides the medical imaging community with an approach to tackle common tasks without requiring paired data or model finetuning.
>
> **The absence of comparable work in the literature speaks to the non-triviality of our contribution.**
>
> **[comment]** *“In addition, the likelihood modeling and posterior sampling components rely on heuristic parameter tuning, without principled justification or adaptive estimation. This makes the framework sensitive to task-specific configurations and limits its reproducibility and general applicability. A deeper analysis or ablation on how the likelihood formulation impacts the final performance would be needed.”*
>
> **[response]** Our DAPS hyperparameters (Appendix A.2, Table 5) follow established settings from prior work and are held constant across all tasks, modalities, and datasets. An extensive hyperparameter sweep would consume substantial computational resources and would miss one of the key advantages of our method; that default parameters generalize well across diverse reconstruction tasks. The only adjusted parameter is τ (noise scale), and we have since found (and updated the submission) that **a universal τ = 0.005** provides a good balance between data fidelity and prior regularization for image restoration and inpainting tasks. **This allows our method to be applied across reconstruction tasks without task-specific tuning.**

---

### Official Review · Reviewer_3fzh · 2025-11-01

**Soundness:** 2
**Presentation:** 3
**Contribution:** 2
**Rating:** 2
**Confidence:** 5

**Summary:**

This paper presents a unified probabilistic framework for 3D brain analysis that bridges the gap between classical and deep learning methods by combining a single, powerful diffusion prior, trained on diverse healthy MRIs , with flexible, task-specific likelihood models at inference time. The authors demonstrate this approach achieves state-of-the-art performance on a range of 3D inverse problems, including super-resolution, bias field correction, inpainting.

**Strengths:**

1. The writing is clear and easy to follow.

**Weaknesses:**

1. The work's conceptual novelty is limited, as it primarily synthesizes established components. The core framework of using score-based diffusion models as "plug-and-play" priors for inverse problems is a well-known concept. The paper's specific choice of sampling algorithm (DAPS) is also directly adopted from prior work.The central claim to novelty is combining this prior with a flexible likelihood (Eq. 11). However, this likelihood function itself is not novel; it represents a data-fitting term ("loss function") whose components are explicitly drawn from prior classical methods. The paper cites previous work for the projection matrix $A$ (modeling alignment, blurring, and downsampling) and for the bias field model.Therefore, the paper's contribution is not a new framework, but rather the application of an existing framework (diffusion priors + posterior sampling) to a specific set of 3D medical imaging tasks. The work consists of "plugging" an established likelihood model (from classical methods like UniRes) into an established diffusion prior framework. This can be viewed as an incremental substitution—swapping a classical prior for a diffusion prior—rather than a fundamental conceptual advance.

2. The diffusion prior is trained only on high-quality, healthy brain scans. This creates a strong bias. This assumption may not be optimal for patients whose "healthy" tissue has been subtly altered or deformed by the pathology, potentially limiting the anatomical fidelity of the inpainted regions.


3. Several key ablation studies are missing: 1. Ablation of Likelihood Components: The main restoration likelihood (Eq. 11) jointly models super-resolution ($A$), image alignment, and bias field correction ($b$). The paper never ablates these components. A crucial missing study would be to compare the full model (Eq. 11) against a simpler version that only performs super-resolution (i.e., without the bias field correction term $b$ and the coordinate descent in Eq. 13). 2. Ablation of the Prior's Training Data: The diffusion prior was trained on a large, diverse cohort of T1w, T2w, and FLAIR images. The paper does not provide an ablation showing how performance changes if the prior is trained on a simpler dataset (e.g., only T1w images). This study would be necessary to justify their significant effort in assembling the multi-modal training cohort. 3. Ablation of Sampler Hyperparameters: The paper uses the DAPS sampling algorithm, which has numerous hyperparameters (e.g., Annealing steps, Diffusion steps, Langevin step number). The paper only provides an ablation for the likelihood precision ($\tau_{y}$).

4. The quantitative comparison for computational cost and model size is missing. The paper provides no data on inference speed or model parameter count for the proposed method or the baselines.

**Questions:**

Same as the weakness part.

---

> ### Author Response · Authors · 2025-11-20
> **Response to reviewer 3fzh (part 1)**
>
> Thank you for your thorough and insightful feedback. We will address each comment accordingly.
>
> **[comment]** *“The central claim to novelty is combining this prior with a flexible likelihood (Eq. 11). However, this likelihood function itself is not novel; it represents a data-fitting term ("loss function") whose components are explicitly drawn from prior classical methods.”*
>
> **[response]** We acknowledge that individual components of our likelihoods draw from prior methods, this is intentional as we aim to incorporate well-established forward models inspired by MRI physics (Equation 11). The novelty lies not in individual components, but in how they are formulated and combined within a unified framework. For inpainting (Equation 14) and refinement (Equation 16), our formulations, whilst simple, are effective for constrained anatomical inpainting and refinement of arbitrary existing reconstructions, addressing previously unsolved problems. Across all tasks and datasets, our method consistently achieves the strongest performance relative to competitive baselines.
>
> To our knowledge, this is the first time such likelihoods have been engineered to integrate with data-driven priors and handle diverse scenarios (low-resolution acquisitions, anisotropic scans, pathological images, refinement of existing methods) without requiring paired training data or access to acquisition parameters. This generalizability across modalities, field strengths, and degradation types within a unified posterior sampling framework represents our core methodological contribution. We also plan to release our code and prior weights, which we believe will provide substantial value to the neuroimaging community.
>
> **[comment]** *“The diffusion prior is trained only on high-quality, healthy brain scans. This creates a strong bias. This assumption may not be optimal for patients whose "healthy" tissue has been subtly altered or deformed by the pathology, potentially limiting the anatomical fidelity of the inpainted regions.”*
>
> **[response]**  Our constrained inpainting formulation (Equation 15) explicitly addresses this concern. The healthy pixel constraint ||Sx_0^(j) - y||^2 forces the reconstruction to preserve the patient's observed healthy tissue, whether perfectly healthy or subtly altered by disease. The inpainted diseased regions are then generated in a way that maintains consistency with this patient-specific tissue, rather than imposing a generic healthy prior, and the iterative optimization ensures anatomical continuity between preserved and inpainted regions.
>
> **[comment]** *“A crucial missing study would be to compare the full model (Eq. 11) against a simpler version that only performs super-resolution (i.e., without the bias field correction term and the coordinate descent in Eq. 13).”*
>
> **[response]** We thank the reviewer for their suggestion. Several images clearly exhibit intensity artifacts (see Figure 2 and Figure 6 in the Appendix), which limits the utility of a comparison with a model that ignores bias-field correction. However we conducted a small-scale analysis on the T1 Clinical cohort  at τ=0.005 with results given in the following table with the best results for each metric highlighted in bold:
> | Method         | MAE     | LPIPS    | SSIM    | PSNR     | VIF      | GMSD    |
> |----------------|---------|----------|---------|----------|----------|---------|
> | Bias field     | **0.0450** | **0.1477** | 0.7501  | **20.9624** | 0.1177 | **0.2165** |
> | No Bias field  | 0.0590  | 0.1502   | **0.7618** | 19.1060  | **0.1227**   | 0.2205  |
>
> For most metrics, the method with bias field adjustment achieves better results. For now, we have only conducted a small scale analysis of bias field effects. However, if the reviewer feels strongly about this point, sufficiently so to consider increasing their score, we would be happy to run an in-depth analysis and include this in the revised submission.
>
> **[comment]** *“2. Ablation of the Prior's Training Data: The diffusion prior was trained on a large, diverse cohort of T1w, T2w, and FLAIR images. The paper does not provide an ablation showing how performance changes if the prior is trained on a simpler dataset (e.g., only T1w images).”*
>
> **[response]** We appreciate this suggestion. However, the purpose of training on diverse contrasts (T1w, T2w, FLAIR) is precisely to enable the generalization we demonstrate. A prior trained only on T1w images would not capture the contrast characteristics needed for T2w and FLAIR reconstruction tasks. We acknowledge that training such a prior requires substantial data collection efforts. To benefit the community, we will release our trained prior weights upon acceptance, allowing researchers to benefit from this diverse training without repeating the data curation process.

---

> > ### Author Response · Authors · 2025-11-20
> > **Response to reviewer 3fzh (part 2)**
> >
> > **[comment]** *“3. Ablation of Sampler Hyperparameters: The paper uses the DAPS sampling algorithm, which has numerous hyperparameters (e.g., Annealing steps, Diffusion steps, Langevin step number). The paper only provides an ablation for the likelihood precision”*
> >
> > **[response]** Our DAPS hyperparameters (Appendix A.2, Table 5) follow established settings from prior work and are held constant across all tasks, modalities, and datasets. An extensive hyperparameter sweep would consume substantial computational resources and would miss one of the key advantages of our method; that default parameters generalize well across diverse reconstruction tasks. We have also updated our results to show that for image restoration and inpainting tasks, setting a universal τ of 0.005 provides a good balance between data fidelity and prior regularization. This allows our method to be applied across these tasks without requiring hyperparameter tuning.
> >
> > **[comment]** *The quantitative comparison for computational cost and model size is missing. The paper provides no data on inference speed or model parameter count for the proposed method or the baselines.”*
> >
> > **[response]** We thank the reviewer for their comment. In the original submission, we compromised detailed methods due to space limitations. We will update the submission to provide these details.

---

### Official Review · Reviewer_hvAu · 2025-11-04

**Soundness:** 3
**Presentation:** 3
**Contribution:** 2
**Rating:** 4
**Confidence:** 3

**Summary:**

This paper proposes a general framework that uses a foundational diffusion prior for refining brain MRI data. To train a reliable diffusion prior, the authors curate a large-scale, artifact‑free brain MRI dataset aggregated from multiple sources. With the trained prior, the method addresses several medical imaging inverse problems and reports strong results across tasks.

**Strengths:**

- The problem is important and highly relevant to the medical imaging community.
- Both the curated dataset and the trained diffusion prior are valuable contributions to the field.
- The paper is generally well presented and easy to follow.
- Experiments across three task setups are thorough and yield meaningful insights.

**Weaknesses:**

- Limited technical novelty. From a task perspective, the medical imaging inverse problems studied here are common in the literature and have been extensively explored, even if prior work may not be as comprehensive as this study. From an algorithmic perspective, using diffusion priors for medical inverse problems has also been investigated,for example, Di‑Fusion (Wu et al., ICLR 2025) and DDM² (Xiang et al., ICLR 2023). While this paper may offer a broader or more unified treatment, the core concepts are not new.
- Insufficient dataset/model detail in the main paper. Given that the primary contributions are (i) dataset curation and (ii) training/validating a foundational diffusion prior, the main paper should include more details and statistics highlighting the dataset’s scope and the model’s practical significance. Currently, key information appears to be deferred to the supplementary materials. It would also be helpful to describe procedures for verifying dataset quality and to compare the curated data with publicly available alternatives.
- Need comparisons with more diffuion-related methods. Empirical comparisons with other medical inverse problem solvers, especially recent diffusion‑based state‑of‑the‑art methods (e.g., Di‑Fusion and DDM²), are not sufficient. Including these benchmarks (on a subset of tasks is fine) would strengthen the evidence.

**Questions:**

The citation style and in‑line references appear inconsistent in places. Could the authors verify and correct the formatting of citations and cross‑references throughout the paper?

---

> ### Author Response · Authors · 2025-11-20
> **Response to reviewer hvAu**
>
> We would like to thank the reviewer for their feedback. We are encouraged to hear that they thought that the problem we are aiming to tackle is highly relevant, that our trained prior was a valuable contribution to the field, and that our experiments were thorough and yielded meaningful results. We will address each concern raised individually here.
>
> **[comment]** *“From an algorithmic perspective, using diffusion priors for medical inverse problems has also been investigated,for example, Di Fusion (Wu et al., ICLR 2025) and DDM² (Xiang et al., ICLR 2023)”*
>
> **[response]** We respectfully disagree, neither Di Fusion (Wu et al., ICLR 2025) nor DDM² (Xiang et al., ICLR 2023) approach MRI denoising or refinement from a principled inverse problem solving perspective and are thus not comparable to our method:
> - Di-Fusion is a conditional generative model which is trained via self-supervision that learns conditional distribution p(x|x’) of one noisy measurement slice x given another x’, whose expectation E[x|x’] equals the clean image y.
> - DDM2 similarly uses the Noise2Noise approach to denoise slice by slice in a self-supervised manner. This estimates the noise level to match inputs to intermediate diffusion states, then a diffusion model is used to generate clean images.
> - **Neither method performs inverse problem with explicit likelihood modelling and posterior sampling.**
> - So whilst these are interesting works they are fundamentally different to our work where we present the first general approach to **combining diffusion model data-driven priors with well-structured likelihoods which incorporate important domain knowledge for MRI brain image restoration.**
>
>
> **[comment]** *“Insufficient dataset/model detail in the main paper.”*
>
> **[response]** Thank you for the suggestion. In the original submission, we tried to find a compromise between providing detailed methods in the main text vs comprehensive results. We have updated the submission to provide more detail on the datasets and the prior training in the main paper.
>
> **[comment]** *“It would also be helpful to describe procedures for verifying dataset quality and to compare the curated data with publicly available alternatives”*
>
> **[response]** Our curated dataset comes from public datasets and is processed by FreeSurfer and Qced. We will clarify this in the main text.
>
>
> **[comment]** *“Empirical comparisons with other medical inverse problem solvers, especially recent diffusion based state of the art methods (e.g., Di Fusion and DDM²)”*
>
> **[response]** As discussed, Di-Fusion and DDM2 perform are not directly comparable to our method as they are self-supervised denoising methods rather than principled inverse problem solvers. However, as requested by the reviewer, we are currently training the Di-Fusion model and will add it as a baseline for the image restoration experiments and update the submission accordingly.
>
>
> **[comment]** *“The citation style and in line references appear inconsistent in places.”*
>
> **[response]** Thank you for drawing this to our attention. We have now fixed the inconsistent reference style in the submission.

---

### Author Response · Authors · 2025-11-25
**Overview of paper changes and clarifications (part 2)**

*[continuation]*

Key Updates in the Revised Submission:

- **Improved performance**: Further training of our diffusion prior has drastically improved our already strong results. On restoration tasks, we now achieve the best performance on all but one IQM (where we rank first or second), with gains of up to 62.6% over the next-best method. We also achieve SOTA performance compared to all baselines on the inpainting tasks.
- **No hyperparameter tuning required**: In our updated submission, we show that all restoration and inpainting tasks perform competitively using the same likelihood precision. This eliminates the need for task-specific tuning, making our method straightforward to apply to new datasets and applications.
- **Additional baseline**: We added Di-Fusion for restoration tasks, as requested by hvAu, and show that our method consistently outperforms it across all datasets.
- **Ablation on bias field**: Following 3fzh’s suggestion, we provide an ablation of Equation 11 with and without the bias field, showing overall improved performance when bias field modeling is included.
- **Details on datasets, training and parameters**: As requested by reviewer hvAu, we moved details on datasets, prior training, and posterior sampling parameters from the appendix to the main text.
- **Computational cost and inference times**: As requested by reviewer 3fzh, we report the parameter counts and inference speed for our method and baselines in Appendix Tables 4 and 5. Whilst our method is on the slower end for inference times, potential strategies to improve sampling time are discussed in Section A.7.

Thank you for your time and feedback. We hope these substantial improvements demonstrate the merit of our work and look forward to the AC’s decision.

---

### Author Response · Authors · 2025-12-02
**Overview of paper changes and clarifications (part 1)**

Dear reviewers and ACs,

We would like to firstly thank the reviewers for their time reviewing our submission.

We were pleased to hear that all reviewers agreed that our submission was well written and easy to follow. All three reviewers note that our method achieves strong results across extensive experiments and hvAu notes that our work is highly relevant and important to medical imaging community.

Although reviewers were unable to respond to our rebuttal before ICLR abruptly closed the rebuttal period on November 28, we believe we have made significant improvements to our submission and clarified several key points. Below, we summarize our clarifications and updates:

- **Our novelty**: We acknowledge the reviewers’ concerns regarding novelty and emphasize that our work is the **first general-purpose image enhancement framework** that unifies data-driven diffusion priors with domain-specific forward models. Our contribution lies not in individual components, but in how they are formulated and combined into a single posterior-sampling framework. Our proposed likelihoods integrate established medical-imaging physics with posterior sampling, bringing together several complementary image-enhancement components in a principled and stable framework. Furthermore, we show that a *single* trained prior outperforms task-specific baselines across diverse inverse problems, modalities, and datasets. By releasing this prior and its accompanying likelihood formulations, **we provide the community with a practical, generalizable tool** that requires no paired data or model finetuning.

- **Comparative methods**: Reviewer hvAu incorrectly compared our method to Di-Fusion and DDM², which are self-supervised denoising frameworks, whereas our method performs **principled posterior sampling**. Reviewer DDwk suggested prior works have applied similar formulations to medical tasks but did not provide any references. Our literature review found no prior work specifically addressing general MRI enhancement via posterior sampling with diffusion priors as we propose.

- **Posterior parameters and ablations**:  Reviewers 3fzh and DDwk raised concerns about the lack of ablations on the posterior hyperparameters and suggested that they require heuristic tuning. We would like to clarify that our DAPS hyperparameters follow established settings from prior work and are held constant across all tasks, modalities, and datasets. An ablation of these parameters would counter one of the key benefits of our work; demonstrating that a single prior and set of posterior settings generalize robustly and successfully across different likelihoods and datasets. This is precisely what enables our method to serve as a general and reliable framework rather than a collection of task-specific methods.

- **Training prior on healthy data**: Reviewer 3fzh raised concerns about our prior being trained solely on healthy data. This is intentional: our goal is to learn a prior that models healthy, 1 mm isotropic, artifact free brain scans and can be applied to a range of tasks. For inpainting of pathological regions, our healthy pixel constraint forces the reconstruction to preserve the patient's observed healthy tissue with the inpainted diseased regions then being generated in a way that maintains consistency with this patient-specific tissue.


- **Need for T1w only model**: Reviewer 3fzh asked why we did not compare our prior to a T1w only model. Training on diverse contrasts (T1w, T2w, FLAIR) is precisely to enable the generalization we demonstrate. A T1w-only prior would not capture the contrast characteristics needed for T2w and FLAIR reconstruction tasks. To ensure the community can fully benefit from the considerable data collection and curation underlying our work, we plan to release our trained prior weights.

*[continued in next comment]*

---

### Meta-Review · Area_Chair_NbmP · 2025-12-29

**Summary:**

This paper proposes a framework for solving 3D brain MRI inverse problems using foundation diffusion priors trained on clean MRI data. Extensive numerical experiments demonstrate consistent improvements in image quality over baseline methods.

The primary concern raised by all reviewers is the limited novelty of the approach. The use of diffusion models as priors for inverse problems in medical imaging, including MRI is well established, and the paper adopts the known DAPS algorithm as the main solver. Reviewers noted that, while the paper may provide a broader or more unified treatment, the core ideas themselves are not new.

During the rebuttal, the authors clarified that their main contribution is positioning the method as a first general-purpose image enhancement framework. However, this clarification does not appear to sufficiently address the reviewers’ concerns regarding novelty.

**Reviewer Concerns:**

During the rebuttal, the authors conducted additional small-scale experiments as requested by the reviewers, while indicating that further experimental results would be left for future work.
The primary concern remains the limited novelty of the paper, as the reviewers were unable to clearly identify a core novel contribution. The authors’ responses mainly emphasize the unification of existing components into a single framework, which does not appear to directly address the reviewers’ central concern regarding novelty.

**Reviewer Scores:**

The reviewers’ initial scores were low (4, 2, and 2), primarily due to concerns about novelty. As these concerns were not fully addressed during the rebuttal, I do not expect the ratings to change significantly.

---

### Decision · Program_Chairs · 2026-01-26

Reject